



# A test of the ability of current bulk optical models to represent the radiative properties of cirrus cloud across the mid- and far-infrared

Richard J. Bantges[1,2] , Helen E. Brindley[1,2] , Jonathan E. Murray[2] , Alan E. Last[2] , Cathryn Fox[3] , Stuart Fox[3] , Chawn Harlow[3] , Sebastian J. O'Shea[4], Keith N. Bower[4], Bryan A.  Baum[5], Ping Yang[6], and Juliet C. Pickering[2]

[1]National Centre for Earth Observation, Imperial College London, UK
[2]Physics Department, Imperial College London, UK
[3]Met Office, UK
[4]University of Manchester, UK
[5]Science and Technology Corporation, Madison, USA
[6]Department of Atmospheric Sciences, Texas A&M University, USA

*Correspondence to*: Richard J. Bantges (r.bantges@imperial.ac.uk)

**Abstract.** Measurements of mid- to far-infrared nadir radiances obtained from the UK Facility for Airborne Atmospheric Measurements (FAAM) BAe-146 aircraft during the Cirrus Coupled Cloud-Radiation Experiment (CIRCCREX) are used to assess the performance of various ice cloud bulk optical (single-scattering) property models. Through use of a minimisation approach, we find that the simulations can reproduce the observed spectra in the mid-infrared to within measurement uncertainty but are unable to simultaneously match the observations over the far-infrared frequency range.  When both mid and far-infrared observations are used to minimise residuals, first order estimates of the flux differences between the best performing simulations and observations indicate a strong compensation effect between the mid and far infrared such that the absolute broadband difference is < 0.7 W m$^{-2}$.  However, simply matching the spectra using the mid-infrared observations in isolation leads to substantially larger discrepancies, with absolute differences reaching ~ 1.8 W m$^{-2}$. These results highlight the benefit of far infrared observations for better constraining retrievals of cirrus cloud properties and their radiative impact, and provide guidance for the development of more realistic ice cloud optical models.

## 1 Introduction

The role of ice clouds (e.g. cirrus) in determining the radiative balance of the Earth and its atmosphere is particularly complex and uncertain [e.g. Baran et al., 2014b; Yang et al., 2015].  The net radiative effect of cirrus is dependent upon its microphysical properties, vertical position and extent, and the geographical location of the cloud [e.g. Heymsfield et al., 2013; Hong and Liu, 2015]. Key microphysical parameters include ice particle habit, particle size distribution (PSD), and morphology such as aggregation, roughness and concavity [Zhang et al., 1999; Baran, 2012; Yang et al., 2012; Baum et al., 2014]. However, these parameters vary both temporally and spatially, and are dependent on the changes in temperature, humidity, and meteorological environment that the cloud experiences [Yang et al., 2012; Voigt et al., 2017].



To accurately predict the radiative effect of cirrus in Global Climate and Numerical Weather Prediction models the interaction of radiation with the ice particles that make up the cloud must be known. This relationship is reliant on knowledge of cirrus optical properties which are currently poorly constrained [Baran, 2012]. Nevertheless, several cirrus bulk optical (single-

scattering) property models have been developed using in-situ observations and databases of ice single scattering properties (SSPs), which aim to include realistic physical representations of ice particles, including habit, aggregation and roughness [Yang et al., 2008, 2012; Baran et al., 2014a,b; Baum et al., 2014].

A key test for these cirrus models is whether, when constrained by suitable in-situ observations, they are able to replicate simultaneous radiance observations across the electromagnetic spectrum. Previous assessments of model consistency with

observations in the solar and mid-infrared (MIR: typically defined as wavenumbers from ~600 to 2500 cm-1) [e.g. Baran and Francis, 2004; Baum et al., 2014; Platnick et al., 2017; Loeb et al., 2018; Yang et al., 2018] have shown reasonable agreement, however there have been very few studies performed in the far infra-red (FIR: defined here as wavenumbers $< 600$ cm$^{-1}$). It is particularly important to validate cirrus models in the FIR as, not only does this region contribute of the order 50 % of the outgoing longwave radiation in the global mean [Harries et al., 2008], but theoretical studies have also indicated that ice

particle optical properties are highly sensitive to radiation in this region [e.g. Maestri and Rizzi, 2003; Baran, 2005, 2007]. In particular, Kuo et al. [2017] show that scattering by cirrus clouds in the far-infrared regime contributes substantially to longwave radiation observed at the top of the atmosphere (TOA). The effect of neglecting longwave scattering is comparable to the effect associated with doubling CO2. Moreover, it has been demonstrated that the addition of a few FIR channels to the MIR channels typically used in satellite retrieval algorithms could significantly improve our ability to retrieve ice cloud

radiative properties [Libois and Blanchet, 2017].

However, compared to the well-studied MIR, spectrally resolved global observations in the FIR are unavailable. There are some measurements available, however, from focused field campaigns. Maestri et al. [2014] analysed downwelling FIR spectral radiances in the presence of cirrus as recorded by the REFIR-PAD instrument [Bianchini and Palchetti, 2008] at the ground-based Testa Grigia station (Italy), 3500 m above sea level, during the Earth COoling by WAter vapouR emission

(ECOWAR) campaign. Retrieved cloud properties from the MIR were used to simulate radiances in the region 250-1100 cm$^{-1}$. Unsurprisingly these showed excellent agreement with the observations in the MIR window from 820-960 cm$^{-1}$ but large residuals were apparent in the FIR from 330-600 cm$^{-1}$. More recent work by Palchetti et al. [2016] and di Natale et al. [2017] suggests that simulated and observed downwelling radiances at the surface can be reconciled within uncertainties across much of the FIR and MIR range if temperature, water vapour and cirrus properties are simultaneously retrieved from the

observations.

Maestri et al. [2014] also discussed the advantages of a 'view-from-above' (satellite or aircraft) experimental configuration for the study of FIR cirrus optical properties. A further benefit of aircraft campaigns for performing model validation exercises is that in principle the radiative signature of the clouds can be recorded closely in time with in-situ measurements of the cloud microphysics, allowing them to be linked directly [Baran and Francis, 2004; Maestri, 2005]. However, to the best of our

knowledge, the only published study based on simultaneous aircraft observations of MIR and FIR radiances in the presence of



cirrus is that of Cox et al. [2010]. This study concluded that simulations were not able to consistently reproduce observed spectral radiances across the infrared and were particularly poor in the FIR region 330-600 cm$^{-1}$, where the analysis was hampered by the presence of large uncertainties in the atmospheric state and a lack of instrumentation capable of measuring small ice particles (diameter < 20 μm).

Here we make use of upwelling radiances recorded above cirrus across both the FIR and MIR spectral regions simultaneously during the Cirrus Coupled Cloud-Radiation Experiment (CIRCCREX), a flight campaign using the Facility for Airborne Atmospheric Measurements (FAAM) British Aerospace (BAe) 146 aircraft based out of Prestwick, Scotland. We focus on flight B895 which took place over the North Sea to the north east of Scotland on the 13th March 2015 and test, for the first time, the ability of the bulk optical models for cirrus developed by Yang et al. [2013] and Baum et al. [2014] to reproduce

radiance observations across the infrared including the FIR. The paper is organised as follows: section 2 provides a summary of the flight and the measurements obtained, along with details of how suitable case studies were identified. Section 3 describes the methodology applied to simulate the observed FIR and MIR spectra and the performance criteria for assessing the "best" matches between observations and simulations. The results are presented in section 4 with conclusions from the study drawn in section 5.

## 2. Observational data

### 2.1 Flight B895 overview

The FAAM flight B895 left Prestwick in the UK on the 13th March 2015, and overflew a decaying band of cirrus cloud associated with an occluded front. The main objective of the flight was to characterise both the cirrus cloud microphysics and their associated IR radiative signatures. To achieve this aim, the aircraft performed a series of straight and level runs (SLRs)

above the cloud followed by a descent into the cloud deck where a further series of SLRs were performed at varying levels within the cloud. Figures 1(a) and (b) show the flight track and altitude of the aircraft as a function of time, respectively.

### 2.2 Instrumentation and measurements

The FAAM aircraft was fitted with a 'cloud-radiation' suite of instruments which included radiation sensors, cloud microphysical probes and a lidar. The radiation instruments included two Fourier transform spectrometers: the Tropospheric

Airborne Fourier Transform Spectrometer (TAFTS, Canas et al. 1997) with nominal spectral coverage from 80 cm$^{-1}$ to 600 cm$^{-1}$ (two channels: LW from 80 cm$^{-1}$ to 300 cm$^{-1}$, and SW from 320 cm$^{-1}$ to 600 cm$^{-1}$) operating at 0.12 cm$^{-1}$ resolution, and the Airborne Research Interferometer Evaluation System (ARIES, Wilson et al. 1999) with nominal spectral coverage from 550 cm$^{-1}$ to 3000 cm$^{-1}$ (two channels: LW from 550 cm$^{-1}$ to 1800 cm$^{-1}$ and SW from 1700 cm$^{-1}$ to 3000 cm$^{-1}$) at 1 cm$^{-1}$ resolution. The cloud microphysics were measured using a series of probes that included a 2DS, a 3 View-Cloud Particle

Imager (3V-CPI), CIP 100 and a holographic cloud probe (HALOHolo), [Lawson et al., 2006; O'Shea et al., 2016]. The 2DS and 3V-CPI are capable of sampling particle sizes from 10 to 1280 μm, the CIP 100 from 100 to 6400 μm and the HALOHolo



from 6 μm to 1 cm. The lidar installed on the aircraft was a Leosphere ALS450 355nm elastic backscatter lidar [Marenco, 2010] which provided cloud vertical profile information and ice volume extinction profiles at 355 nm from the range-corrected backscatter profiles following the two-stage process of Marenco et al. [2011]. Vertical profiles of particle extinction coefficient were estimated [Fox et al., 2019] from which the cloud optical depth at 355 nm, ($\tau_{355}$), was derived. Finally, along with the standard aircraft positioning sensors, temperature and humidity sensors, the aircraft was also equipped with the Airborne Vertical Atmospheric Profiling System (AVAPS) [Vaisala, 1999] which launched RD94 dropsondes [Vaisala, 2010] at various stages in the flight to enable characterisation of the atmospheric column below the aircraft.

**2.3 Identification of suitable case studies**

The driving factor in selecting periods for analysis was the availability of near-simultaneous TAFTS and ARIES nadir radiance spectra from SLRs above the cloud. Ideally, simulation of radiance spectra equivalent to those observed at the aircraft level above the cirrus cloud requires knowledge of the cloud microphysical and optical properties, its vertical location and the atmospheric profile from the surface to the aircraft. Hence, additional selection criteria were that cirrus could be clearly identified and characterised by the lidar observations and that the atmospheric state below the aircraft was well-characterised by dropsonde measurements.

Preliminary examination of the available data from the three above-cloud SLRs suggested that there were two periods which satisfied these requirements; one during SLR 1 from 09:48:39 to 09:49:51 UTC and another during SLR 3 from 10:16:22 to 10:17:24 UTC. Unfortunately, ARIES data from the period identified in SLR 3 were found to have deficiencies [S Fox, pers. comm. 2019], and considered to have insufficient quality for the purpose of this study. Consequently, only those data from the period during SLR 1 were considered for further investigation. Three sets of radiance observations were identified from within SLR 1: the times of these observations along with a summary of the conditions at the aircraft are summarised in Table 1.

For each of the three cases identified, only a single TAFTS spectrum was available; however, owing to the relatively high temporal sampling frequency of the ARIES instrument, eight ARIES spectra are available within ± 1 s of the TAFTS acquisition time. To provide an indication of the scene variability within the two second period around the TAFTS measurement, the mean and associated standard deviation are calculated for the eight ARIES radiance spectra. The mean ARIES and the associated TAFTS spectra are then converted into equivalent brightness temperature spectra. These are shown for all three cases in Fig. 2. In the MIR, signatures of cirrus are most apparent in the main atmospheric window (~760-1000, 1080-1250 cm$^{-1}$) as indicated in Fig. 2 (c). At FIR frequencies, the atmosphere is less transparent because water vapour absorption fills in many spectral lines, but there are a series of semi-transparent, so called "micro-windows" as shown in Fig. 2 (b), that clearly show the variation in the cirrus properties. For the lowest frequencies in the FIR, the atmosphere is comparatively opaque with generally a much lower sensitivity (less than 1 K) to the cirrus with the exception of a few wavenumber regions (e.g. ~110 cm$^{-1}$, 218 to 221cm$^{-1}$, 240 cm$^{-1}$, etc.). It is useful to note that there is a variation in the ordering of the three cases between the different spectral regions. For example, at around 110 cm$^{-1}$ the maximum brightness temperature is observed for case C followed by B and then A; case B shows the highest values in micro-window 3 (around 410 cm$^{-1}$)





followed by A and then C; in the MIR window regions the largest values are for case A, followed by B and then C. The variation in the relative ordering of the individual cases indicates a frequency dependent sensitivity to cirrus properties: this sensitivity to the ice particle size and habit is discussed further in section 4.2.

Figure 3(a) shows the lidar extinction profiles for the three cases (A-C). Panels (b) and (c) show the atmospheric temperature and water vapour mixing ratio profiles obtained from the two dropsondes deployed nearest to the time of the selected radiance

observations at 09:47:48 UTC (ds1) and 09:50:52 (ds2). The aircraft altitude is shown by the dashed red line. The extinction profiles derived from the lidar measurements indicate a band of cirrus was located between approximately 6 and 9 km in altitude with no evidence of any underlying cloud. Integration of these extinction profiles for the three cases, from 6 to 9 km, indicates that $\tau_{355}$ ranged from approximately 0.53 to 0.59. These clouds would therefore typically have been classified as optically thin [e.g. Dessler and Yang, 2003]. Examination of ds1 and ds2 demonstrates that between their deployments the

atmospheric state remained relatively stable, with a noticeably dry layer between approximately 3 and 5 km evident in both water vapour profiles. There is a degree of variability in the temperature profile around 4 km, and in the water vapour mixing ratio below 3 km. The impact of this variability on the simulated radiance spectra is discussed in section 4.

## 3. Simulation methodology

The simulation approach makes use of two radiative transfer models, the Line by Line Radiative Transfer Model (LBLRTM, [Clough et al., 2005]) and the Line by Line Discrete Ordinates (LBLDIS v3.0, [Turner, 2005]) code, in conjunction with a representation of the atmospheric state including cloud location and microphysics, expressed in terms of optical depth and effective radius.

For all three cases simulated the aircraft was overflying ocean. The radiative temperature of the ocean surface was assumed to

equal the temperature measured at the lowest altitude from the closest dropsonde in time to the radiance observation. These values were compared to collocated European Centre for Medium Range Weather Forecasts (ECMWF) Interim Reanalysis (ERA-I; [Dee et al., 2011]) surface skin temperatures and were found in all cases to agree to within 0.1 K. The ocean surface spectral emissivity was defined using the wind-speed dependent model from Masuda et al. [1988], with estimates of the wind speed obtained from the nearest available ERA-I value. The atmospheric column beneath the aircraft was divided into 0.1 km

thick layers, providing 93 layers from the surface to 9.3 km, with an additional layer closest to the aircraft varying in thickness depending upon the aircraft altitude (between 0.085 and 0.091 km). Temperature and water vapour concentrations for each layer were obtained by interpolating the corresponding dropsonde measurements onto the 0.1 km vertical grid, with the exception of the layer closest to the aircraft which was prescribed by the onboard measurements. In the absence of direct measurements, concentrations of carbon dioxide and minor trace gases were obtained from a standard Mid-Latitude Winter

(MLW) atmospheric profile [Anderson et al., 1986] and scaled to present day concentrations. Ozone concentrations were obtained from collocated ERA-I data and interpolated to the required vertical resolution.



The quantities describing the surface and atmospheric column were then input into LBLRTM to calculate the optical depths for every layer in the atmosphere between the surface and the aircraft over the spectral range 105 cm$^{-1}$ to 1600 cm$^{-1}$. The latest version currently available, LBLRTM v12.8, was used with the molecular absorption defined by the Atmospheric and Environmental Research (AER) v3.6 spectral line database which is based on the HITRAN 2012 database [Rothman et al., 2013]. The water vapour continuum was defined by the recently released MlawerTobin_Clough-Kneizys-Davies (MT_CKD) 3.2 [Mlawer et al., 2019].

The wavelength-dependent LBLRTM-derived layer optical depths were then passed to LBLDIS, which takes into account scattering by particles in the cloud layer via the Discrete Ordinate Radiative Transfer (DISORT) code [Stamnes et al., 2000]. In this study the bulk optical properties used to simulate cloudy radiances were those provided by Baum et al. [2014]. This parameterisation consists of three databases that are based on different ice particle habits: solid columns only (SC); the aggregate of solid columns only (ASC); and a general habit mixture (GHM) that incorporates plates, droxtals, hollow and solid columns, hollow and solid bullet rosettes, an aggregate of solid columns and a small/large aggregate of plates. Further details on the geometries of these habits are given in Baum et al. [2011] while the method by which the parameterisation was built is reported in Baum et al. [2005a, 2005b, 2007]. Each database contains the SSPs expressed as a function of wavelength (0.2 to 100 μm) for a range of PSDs, assuming a gamma distribution (see Heymsfield et al. [2013]), with particle effective radius ($r_{eff}$) ranging from 5 to 60 μm. LBLDIS also requires the cloud height, $r_{eff}$, and optical depth for each cloud layer as input. Based on these input parameters, LBLDIS is used to simulate radiances over a wavenumber range and with a spectral resolution set by the user.

Appropriate instrument apodisation functions were then applied to the simulated radiance spectra. To ensure that this process did not introduce errors, the simulations were performed at 0.01 cm$^{-1}$ resolution. Once the instrument apodisation functions had been applied, the simulated spectra were interpolated onto the same wavenumber scale as the observations, facilitating direct comparison. A schematic summarising the simulation methodology is provided in Fig. 4.

## 4. Results

We separate the results into three sub-sections. The first section describes the initial efforts to simulate the observed spectra utilising the best available information on the cirrus cloud properties. The following section describes the minimisation methods adopted to find the best agreement between the simulated and observed spectra and the results using these methods are presented in the final section.

### 4.1 Initial simulation

A single aircraft is not able to simultaneously measure in-situ cloud properties and above cloud radiance spectra. Here, the in-situ cloud microphysical measurements were obtained over one hour after the radiation observations were obtained (Fig. 1). While the cloud vertical structure remained relatively constant during Cases A-C (Fig. 3a), the lidar observations indicated





significant variability in the geometrical thickness of the observed cloud during the three SLRs as a whole. Examination of the available in-situ cloud microphysical properties [O'Shea et al., 2016] also indicated a high temporal (and therefore implied spatial) variation in the cloud PSD. These issues, combined with the knowledge that the cloud was decaying over time, suggested that it would be difficult to associate a particular observed PSD with any confidence to the radiation measurements. Therefore, an initial simulation for each case was performed for a cloud, separated into layers 0.1 km thick, of vertical extent from 6 to 9 km and taking bulk optical properties from the Baum et al. [2014] ASC ice particle habit model, and assuming $r_{eff}$ = 30 μm. The relative optical depth for each cloud layer was derived from the lidar observations, ensuring the cloud's total optical depth was equal to the appropriate value from Table 1.

Figure 5 shows the results of such a simulation for case A. For reference, the equivalent clear-sky simulation is also shown. The estimated uncertainty associated with the ARIES spectral calibration is approximately 1 K [S Fox, Met. Office, Personal Communication, 2019], while the 1σ variability in the eight ARIES spectra matched to the TAFTS acquisition time is approximately ± 0.5 K. We therefore estimate the total uncertainty on a mean ARIES spectrum as these values added in quadrature, so ~1.1 K. However, the simulated brightness temperature spectrum overestimates the observed ARIES values in the main atmospheric window by around 5 K, with a similar overestimate seen relative to the TAFTS measurements in the FIR micro-windows. Conversely, the level of agreement in the strong $CO_2$ absorption band from 650 to 700 cm$^{-1}$ (excluding the spike in the centre of the band at 667 cm$^{-1}$), indicates that the temperature of the uppermost layer in the simulation (i.e. close to the aircraft) is well represented in the temperature profile used in the simulation. These differences point to issues with the cloud properties used in the initial simulation, especially given that interchanging dropsondes 1 and 2 has a comparatively small impact on the simulated spectrum.

## 4.2 Achieving an improved simulation-observation fit

It is evident from Fig. 5 that the initial choice of cloud parameters used to simulate the observed radiance were sub-optimal. The two key parameters required to define the microphysical properties of an ice cloud are $r_{eff}$ and the ice particle habit. The sensitivity of the mid-infrared to ice particle size has been known for many years [e.g. Bantges *et al.*, 1999] and more recently sensitivity studies extending into the far-infrared have been performed [e.g. Yang, 2003; Yang et al., 2013]. Figure 6 shows the results of a series of simulations performed to examine the impact of $r_{eff}$, ice particle habit and optical depth; the simulated radiances are developed using the Baum et al. [2014] models. The results indicate that there are a wide range of spectral regions that demonstrate sensitivity to size from approximately 300 to 500 cm$^{-1}$, 750 to 850 cm$^{-1}$ and 950 to 1250 cm$^{-1}$. Note however that the ordering of the differences changes around 400 cm$^{-1}$ where the largest $r_{eff}$ no longer shows the greatest sensitivity. In contrast, there are spectral regions that exhibit sensitivity primarily to ice particle habit. Differences between the ASC and SC model are greatest around 550 cm$^{-1}$, while they are greatest at around 400 cm$^{-1}$ for ASC and GHM differences. Sensitivity to increasing optical depth is broadly similar across the MIR and FIR from 400 to 1400 cm$^{-1}$, but drops off rapidly at wavenumbers lower than 400 cm$^{-1}$ due to the increasing effect of strong water vapour absorption and the overall reduction in radiative energy at these frequencies.



With use of this information, a scheme was developed to minimise the differences between the simulated and observed spectra in regions showing particular sensitivity to $r_{eff}$ and habit. Four wavenumber regions in the MIR and four in the FIR were used; 775, 850, 900 and 1200 cm$^{-1}$ in the MIR and 365, 410, 450 and 497 cm$^{-1}$ in the FIR. The ultimate goal was to investigate whether there was any combination of parameters that could fit the observations across the MIR and FIR simultaneously given

measurement uncertainties.

To facilitate the minimisation a series of simulations were performed for a range of $r_{eff}$ and $\tau$ for the three ice particle habits in the Baum et al. [2014] database. Table 2 provides a summary of the ranges covered for each variable. These values were chosen, based on initial simulation attempts to match the observed spectrum, to produce a range of simulations that encompassed the observations and their associated uncertainties.

The simulations were compiled into a database ordered by $\tau_{355}$, $r_{eff}$ and ice particle habit. Four different approaches were then adopted to identify the simulation that most closely matched the observations for each case. The first approach comprised two stages. The initial stage identified those simulated radiance spectra that agreed to within the measurement uncertainties for the ARIES spectrum for all four of the selected MIR channels to create a subset of 'matched' simulated spectra. This subset was then processed to find the simulation that most closely matched the corresponding TAFTS SW spectrum for the four channels

identified in the FIR. The absolute difference between the simulations and observations for each FIR channel was weighted by the observation uncertainty and then summed,

$$x = \sum_{i=1}^{4} \frac{|Rsim_i - Robs_i|}{Rerror_i} \qquad [1]$$

where $Rsim_i$ and $Robs_i$ are the simulated and observed radiances, respectively and $Rerror_i$ is the observation uncertainty for FIR channel $i$. X was then minimised to obtain the 'best' solution.

The second approach also comprised two stages, with the first stage mimicking that of the first approach. In the second step the uncertainty weighting was removed such that the minimum of equation 2 was sought,

$$x' = \sum_{i=1}^{4} |Rsim_i - Robs_i| \qquad [2].$$

The third approach focussed on determining the best agreement that could be obtained using the individual MIR spectral range. This was achieved by minimising the integrated absolute difference between the simulation and observation across the MIR

from 600 to 1400 cm$^{-1}$. For completeness, a fourth approach, similar to the third, but in this case minimising the integrated difference in the FIR from 320 to 540 cm$^{-1}$, was considered to demonstrate whether agreement solely in the FIR could be achieved.

### 4.3 Minimisation results

The combinations of ice particle habit, $r_{eff}$ and $\tau_{355}$ giving the simulated spectra that most closely match the observations using

each of the three minimisation methods are summarised in Table 3. The results for all the methods indicate a need to increase $\tau_{355nm}$ above the lidar-derived value by between 25 to 45 %, somewhat more than would be anticipated given the estimated uncertainty in the lidar values are ±20 %. The deviation may be a consequence of an inconsistency between the optical





properties implicitly assumed when converting the raw lidar measurements to optical depth compared with those used in the simulations here [e.g. Heymsfield et al., 2008; Fox et al., 2019].

Minimisation methods 1 and 2 yield very similar results, both indicating that the GHM habit provides the closest agreement, with the only difference that method 1 suggests a slightly smaller $r_{eff}$ compared to method 2 for Case A. Method 3 demonstrates that using the MIR in isolation gives markedly different values. In this case the results would suggest a cloud comprised of relatively small ASC habit ice particles with an optical depth slightly larger than that indicated by methods 1 and 2. Finally, when matching the FIR observations in isolation (method 4), a cloud with lower optical depth composed of relatively large

GHM habit ice particles is implied.

The spectral differences between the observations and best matching simulations are shown in Fig. 7 for each minimisation method, separated into 3 different spectral regions defined as MIR (600 to 1400 $cm^{-1}$), SW FIR (320 to 540 $cm^{-1}$) and LW FIR (110 to 300 $cm^{-1}$). It is important to note that the large differences consistently found around 667 $cm^{-1}$ are most likely a measurement artefact due to the extremely strong absorption by $CO_2$ around this frequency. This results in measurements

reflecting the temperature of the air inside or very close to the ARIES instrument. Similar effects can be present due to strong water vapour lines and are particularly evident at wavenumbers > 1300 $cm^{-1}$. As evidenced by Fig. 3, cloud signatures are more apparent in the more transparent spectral regions indicated by the "window" labels in Fig. 7(a) and where the atmospheric transmission between the cloud-top and aircraft is close to 1 (Figs. 7(b) and (c)).

Unsurprisingly given the values in Table 2, the difference spectra from methods 1 and 2 lie almost on top of each other across

the full spectral range analysed. Within the MIR (Fig. 7(a)), the differences are within the measurement uncertainties with the exception of those spectral regions most strongly influenced by $CO_2$ and water vapour discussed above. Within the TAFTS LW channel (Fig. 7(c)), the simulations overlie each other since, as shown in Fig. 6, for frequencies lower than around 300 $cm^{-1}$, the models predict a greatly reduced sensitivity to ice particle habit and size. The atmospheric layer above the cloud, in this spectral region is also less transparent, further reducing the observed impact of variations in the ice particle properties.

However, in the SW FIR there are several regions (e.g. 375 $cm^{-1}$, 450 $cm^{-1}$ and 475 $cm^{-1}$) where the differences lie outside of the measurement uncertainty estimates (Fig. 7(b)). Overall, for case A, a total of 1488 simulations were performed. Of these, 739 matched the observed spectrum to within uncertainties in the MIR and 14 matched the observed spectrum in the FIR. However, as implied by Fig. 7, the simulations that matched in the FIR were not contained within the set of simulations that matched in the MIR. This result shows that for the bulk optical models tested, any combination of $\tau_{355}$ and $r_{eff}$ is unable to

match the TAFTS and ARIES observations simultaneously within measurement error. Analysis of cases B and C (not shown) yields similar conclusions.

It is worth reiterating that the difference spectra obtained using method 3 demonstrate that it is possible to achieve excellent agreement in the MIR using MIR observations in isolation (Fig. 7(a)). However, extending the best fitting parameters to the FIR generates large residuals (Fig. 7(b)), while using the FIR in isolation yields closer agreement across the FIR (albeit not

within measurement uncertainty across all micro-windows) but even larger residuals in the MIR. This highlights that the





combined use of MIR and FIR information provides a much tighter constraint on retrieval quality (consistent with Libois and Blanchet, [2017]) and that FIR observations are needed to refine and test bulk ice cloud optical model development.

To assess the energetic impact of misrepresenting the cirrus spectral signature, a first order estimate of the upwelling flux difference between the simulations selected by methods 1-3 and the observations is obtained by multiplying the upwelling
nadir radiances by pi (assuming an approximation to the diffusivity-factor approximation, [Elsasser 1942]) and integrating over selected spectral ranges (Table 4). For completeness, results from all three cases analysed are shown. Table 4 indicates that, in all cases and for all methods there is a compensation effect between the differences seen in the MIR and in the shortwave FIR channel. This compensation means that for Methods 1 and 2, rather large differences (of the order 0.6-1.6 W m$^{-2}$) are seen in the individual bands but these are somewhat masked when looking at the deviation integrated across all three bands. Our
results suggest that the use of information solely from the MIR to estimate the cirrus properties (method 3) can result in sizeable deviations in both the SW FIR and the total flux, exceeding 2.0 W m$^{-2}$ for one of the cases analysed.

**Discussion and Conclusions**

The ability to simulate the radiative signatures of cirrus cloud across the mid- and far-infrared as measured by the ARIES and TAFTS instruments on the FAAM aircraft has been assessed using sophisticated radiative transfer codes in combination with
state-of-the-art cirrus optical property databases. Despite considering a wide range of cloud properties, comparisons between the simulated and observed spectra have shown that it is currently not possible to achieve agreement across the infrared to within the estimated measurement uncertainties for the Baum et al. [2014] bulk optical (single-scattering) property models tested here.

With use of a variety of minimisation approaches, we have shown that restricting the matching frequency range to the mid-
infrared generates multiple solutions that span a wide range of cirrus optical properties. These can give agreement with the observations to within measurement uncertainty in the mid-infrared but result in discrepancies that exceed measurement uncertainty at far-infrared frequencies. While no solutions are able to capture the observed mid and far-infrared behaviour to within uncertainties, the combined use of the mid- and far-infrared gives the tightest constraint on optical model parameters thus enabling the identification of the most representative optical properties for the observed cloud.

First order analysis of the longwave energetic flux errors that result from deficiencies in the simulations' ability to represent the radiative properties of the observed thin cirrus demonstrate that these can be significant, reaching of the order of 1 to 2 Wm$^{-2}$ across the far-infrared. Interestingly, these errors tend to compensate when summed across the infrared as a whole and would not be very apparent in broadband flux measurements. This illustrates the need for spectral information, since spectrally dependent errors would be expected to translate into errors in the vertical profile of heating [e.g. Clough et al., 1995; Brindley
and Harries, 1998; Turner et al., 2018] potentially affecting cloud and atmospheric dynamics.





It is noted that the ice cloud optical property databases employed in this study do not yet take into account the temperature dependency of the refractive indices of ice [Iwabuchi and Yang, 2011] and that initial studies imply that this may exert a noticeable impact on retrievals of cirrus optical depth and effective radius utilising the MIR and FIR [P Yang, pers. comm., 2019]. A further question relates to the relative sensitivity of the far and mid-infrared regimes to depth within the cloud. To address this, the impact of varying the vertical profile of $r_{eff}$ within the cloud on the simulated spectra was also considered (not shown), but found to have only a minor impact (typically $< 0.05$ K).

Within the Baum et al. [2014] bulk optical models there is also an implicit assumption that the cloud PSD follows a gamma distribution. However, these PSDs were generated from in situ aircraft measurements from a variety of field campaigns [Heymsfield et al., 2013]. How realistic these fitted PSDs may be for the clouds studied here is an open question; however using PSDs generated from the in-situ measurements of the cloud microphysics taken one hour after the radiation measurements does not improve upon the simulation performance. We note again that the high variability of the cloud field also raises questions as to how closely these in-situ measurements represent the cloud sampled by the radiation instruments. Future campaigns should seek to employ a dual platform approach to enable simultaneous in-cloud sampling and above cloud radiative observations.

The European Space Agency recently announced the selection of its ninth Earth Explorer mission, Far-infared Outgoing Radiation Understanding and Monitoring (FORUM), scheduled for launch in the 2025 timeframe. The mission will see the Earth's emitted radiation from 100 to 1600 cm$^{-1}$ measured from sun-synchronous orbit, providing global coverage for at least 4 years. These observations should enable a consistent link to be made between the microphysical properties of cirri and their radiative signatures, ultimately helping to provide an improved representation of cirrus clouds in climate and forecast models, both in terms of the physical processes driving specific types of cirri formation and their associated impact on the Earth's energy budget. Key preparatory steps will be the development of bulk optical property models for ice cloud that show consistency with measurements across the electromagnetic spectrum, necessitating additional airborne observations of cirrus and their associated microphysical properties over a range of cirrus types in addition to the optically thin frontal cloud analysed here.

**Data availability**

Data from the CIRCCREX field campaign can be found at the Natural Environment Research Council's Data Repository for Atmospheric Science and Earth Observation: https://catalogue.ceda.ac.uk/uuid/6ba397d6c8854da19bcced8ea588c1f9.

**Author contributions**

RB performed the simulations and designed and wrote the manuscript. HB provided guidance and focus for the study. JM, AL and JP were responsible for enabling the measurements from the TAFTS instrument as part of the CIRCCREX campaign flight



B895. CF provided some background material. SF and RH provided ARIES and LiDAR data. PY and BB provided ice cloud optical models and guidance. KB and SO provided information on the in-situ microphysical data obtain during the CIRCCREX campaign flight B895.

**Competing interests**

The authors declare that they have no conflict of interest.

**Acknowledgements**

This study was funded by ESA contract number 4000124917. The CIRCCREX campaign and subsequent data analyses were funded by the Natural Environment Research Council (UK) grants NE/K015133/1 and NE/K01515X/1.

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





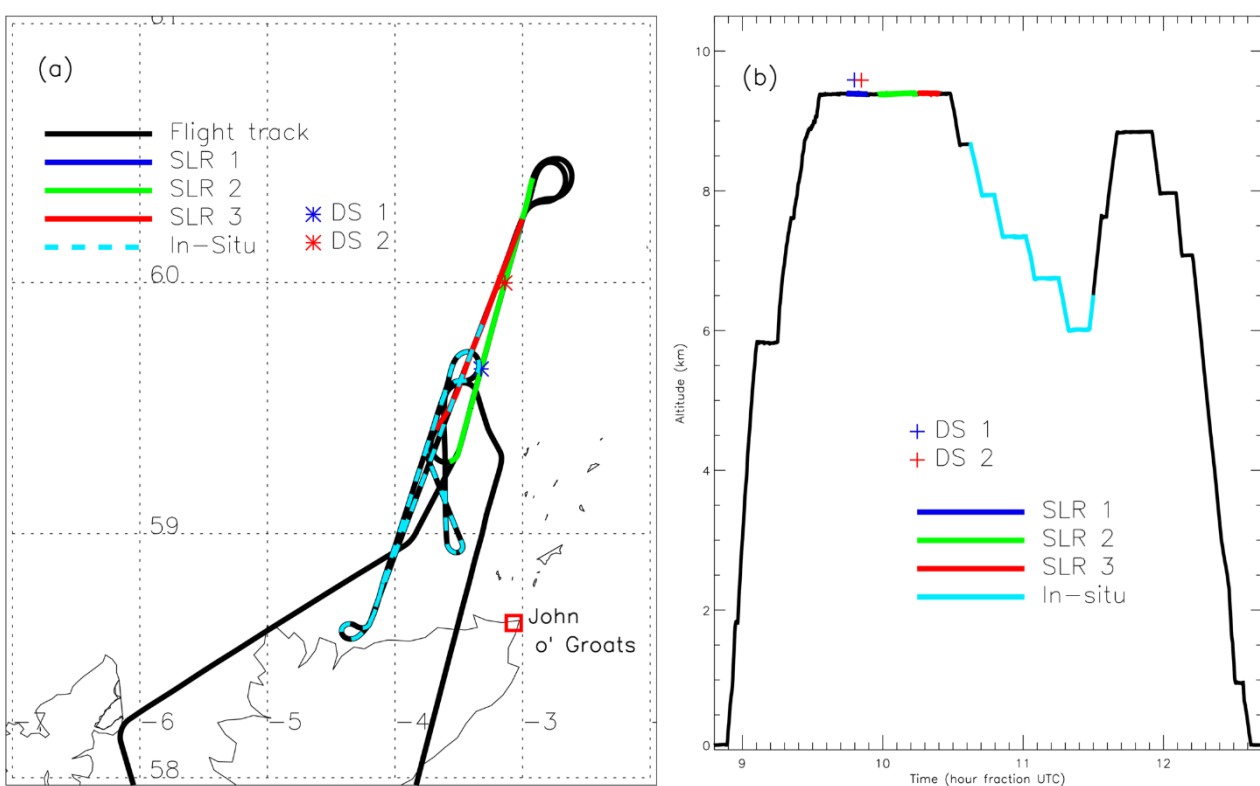

**Figure 1: (a) Flight track for FAAM Flight B895 shown in black. The three SLRs are shown in colour, but note that SLR-1 lies directly beneath SLR-2 and is therefore not visible. The deployment of dropsondes (DS) 1 and 2 are indicated along with the period following SLR 3 when in-situ sampling of the cloud layer was performed. (b) Aircraft altitude as a function of time, with the three SLRs and in-situ measurement phases shown, and the DS 1 and 2 deployment times.**





**Figure 2: Brightness temperature spectra for the three cases A-C. (a): Individual TAFTS LW channel spectra. (b): Individual TAFTS SW channel spectra. (c): For each case, the mean of eight ARIES individual radiance spectra, converted to equivalent brightness temperature spectra. Also shown are a selection of atmospheric micro-window regions in (b), and the two main window regions in (c).**

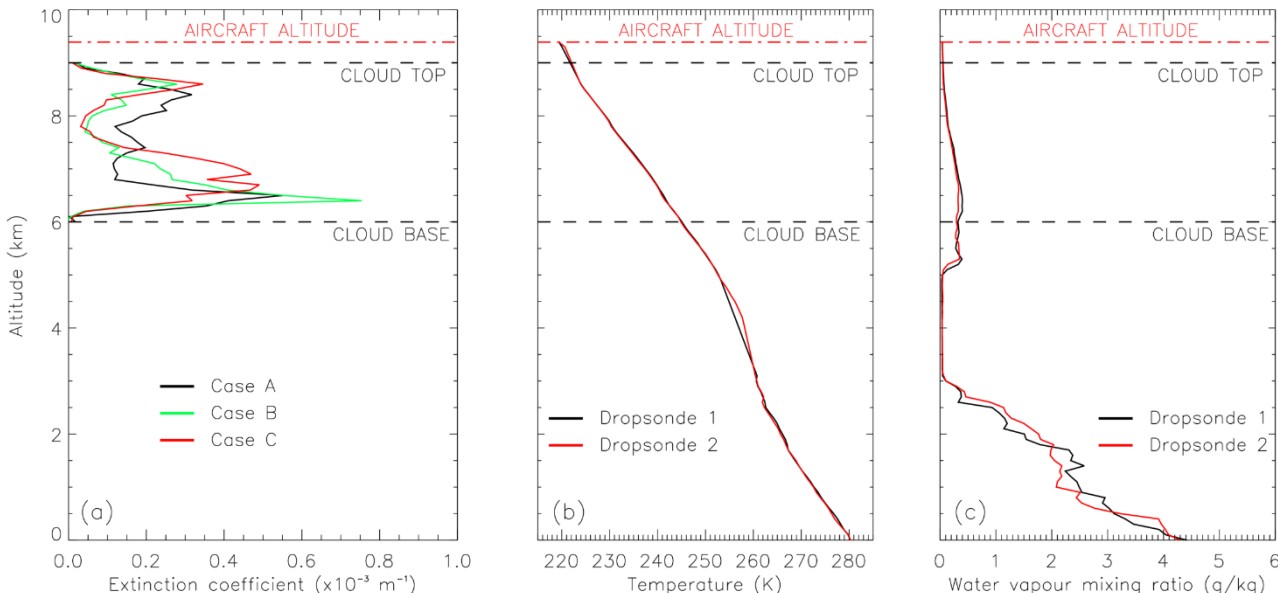

**Figure 3: (a) Lidar extinction coefficient profiles for cases A to C. (b) Dropsonde temperature profiles from the two dropsondes deployed during SLR 1. (c) As (b) but for the water vapour mixing ratio profiles. The altitude of the aircraft and the upper and lower boundaries of the cirrus cloud layer are indicated by the black dashed lines.**

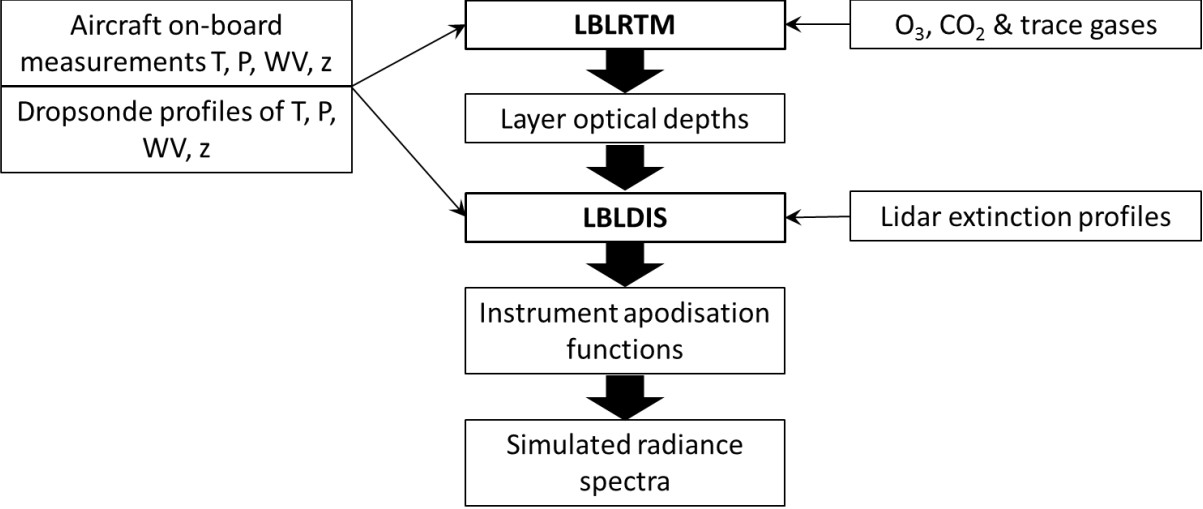

**Figure 4: Schematic of the inputs (T – temperature, P – pressure, WV – water vapour, z – altitude) and steps involved in simulating the radiance spectra observed by the TAFTS and ARIES spectrometers.**





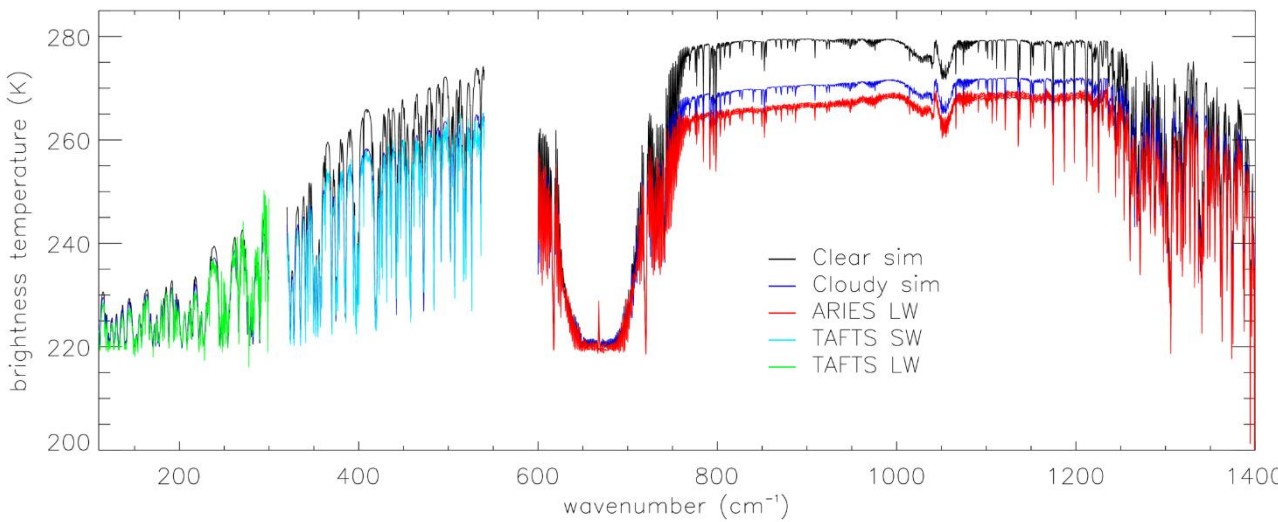


**Figure 5: Simulation for case A assuming a cirrus cloud with $\tau_{355}$ = 0.584, composed of ice particles with $r_{eff}$ = 30 μm, and Baum ASC particle habit. The TAFTS LW and SW channel spectra are each a single spectrum. The ARIES LW channel spectrum plotted is a mean of 8 ARIES observations, and the width of the curve indicates the standard deviation about the mean.**







 **Figure 6: Simulated brightness temperature difference spectra to illustrate where there is sensitivity to $r_{eff}$ (a), particle habit (b) and $\tau_{355}$ (c). Panel (a) shows differences relative to a simulation using the ASC model assuming $\tau_{355} = 1.46$ and $r_{eff} = 10$ μm. Panel (b)**





shows differences relative to a simulation using the ASC model assuming $\tau_{355} = 0.87$ and $r_{eff} = 30$ μm. Panel (c) shows differences relative to a simulation using the ASC model assuming $\tau_{355} = 0.5$ and $r_{eff} = 30$ μm.






**Figure 7: Simulation minus observation difference spectra for case A for each minimisation approach, for the (a) MIR, (b) SW FIR and (c) LW FIR frequency ranges. The transmission of the atmosphere (derived from a simulation) between the cloud top and the aircraft level is also shown in the lower two panels.**





| CASE | TAFTS obs. time (UTC) & *sSM* | ARIES obs. time (sSM) | LiDAR obs. time (sSM) | $\tau_{355}$ | Pressure at Aircraft (hPa) | Temperature at Aircraft (K) | Altitude at Aircraft (km) |
|---|---|---|---|---|---|---|---|
| A | 09:48:39 *35319* | 35318.3, 35318.8, 35319.3, 35319.8 | 35315 | 0.584 | 286.864 | 219.407 | 9.391 |
| B | 09:49:14 *35354* | 35353.0, 35353.5, 35354.0 35354.5 | 35347 | 0.531 | 287.001 | 220.224 | 9.387 |
| C | 09:49:51 *35391* | 35390.2, 35390.7, 35391.2 35391.7 | 35381 | 0.592 | 286.995 | 219.692 | 9.387 |

**Table 1: Summary of the three cases (A,B and C) identified for use in this study, detailing the time of TAFTS, ARIES and LIDAR observations, along with the key variables from the on-board aircraft instrumentation obtained at the time of the TAFTS observations. Note that the TAFTS observation (obs.) times represent the UTC time (HH:MM:SS) and the equivalent seconds Since Midnight (sSM) times. Also, note that there are 2 ARIES spectra (forward and reverse scan directions of the interferometer) associated with each ARIES observation time. The LiDAR time indicates the start of a 10 second period over which the data are**
**averaged.**





| Case | Ice particle habit | $\tau_{355}$ | $r_{eff}$ (µm) |
|:---:|:---:|:---|:---|
| A | ASC, SC & GHM | 0.5 to 0.8 at 0.02 resolution | 20 to 40 at 2 resolution, and 45 |
| B | ASC, SC & GHM | 0.87 to 1.03 at 0.01 resolution | 20, 22, 25 to 40 at 1 resolution, and 45 |
| C | ASC, SC & GHM | 0.96 to 1.14 at 0.01 resolution | 20, 22, 25 to 40 at 1 resolution, and 45 |

**Table 2: Summary of the variation in the cloud optical properties for which radiance spectra were simulated for cases A, B and C.**



| Method (approach) | Ice particle habit (A,B,C) | $r_{eff}$ (A,B,C) | $\tau_{355}$ (A,B,C) |
|---|---|---|---|
| 1 | GHM, GHM, GHM | 34, 38, 45 | 0.82, 0.93, 1.07 |
| 2 | GHM, GHM, GHM | 40, 38, 45 | 0.82, 0.93, 1.07 |
| 3 | ASC, ASC, ASC | 28, 31, 31 | 0.86, 0.97, 1.09 |
| 4 | GHM, GHM, GHM | 40, 45, 45 | 0.72, 0.86, 0.96 |

**Table 3: Summary of the cloud optical properties that yield the closest agreement between the simulated and observed spectra for the four different minimisation approaches.**



| Approach number | 600 to 1400 cm⁻¹ (Wm⁻²) | | | 320 to 540 cm⁻¹ (Wm⁻²) | | | 110 to 300 cm⁻¹ (Wm⁻²) | | | TOTAL (Wm⁻²) | | |
|---|---|---|---|---|---|---|---|---|---|---|---|---|
| | A | B | C | A | B | C | A | B | C | A | B | C |
| 1 | 1.10 | 0.92 | 0.69 | -0.64 | -1.55 | -0.88 | -0.08 | -0.01 | 0.19 | 0.38 | -0.64 | 0.00 |
| 2 | 0.84 | 0.92 | 0.69 | -0.64 | -1.55 | -0.88 | -0.08 | -0.01 | 0.19 | 0.12 | -0.64 | 0.00 |
| 3 | 0.17 | 0.22 | 0.31 | -1.15 | -2.02 | -1.37 | -0.09 | -0.02 | 0.19 | -1.07 | -1.82 | -0.87 |

**Table 4: Integrated, simulation minus observation, flux differences for selected spectral ranges (MIR, SW FIR and LW FIR) and the total, for the best matching spectra obtained using the first three different minimisation methods, for the three case studies. Note, results from method 4 are not included as we do not anticipate future observations to be restricted to only FIR frequencies.**