# Peer review of "A test of the ability of current bulk optical models to represent the radiative properties of cirrus cloud across the mid- and far-infrared"

_Atmospheric Chemistry and Physics, 2019_

## Referee Comment (RC1) · Quentin Libois (Referee) · 14 Apr 2020

Review of « A test of the ability of current bulk optical models to represent the radiative properties of cirrus cloud across the mid- and far-infrared », by Richard J. Bantges et al.

**General comments**

This paper investigates the capability of a cloud single scattering properties (SSPs) database (taken from Baum et al., 2014) to explain downlooking airborne radiance observations above cirri throughout the mid- and far-infrared taken with the ARIES and TAFTS instruments during the CIRCCREX experiment in March 2015. Short flight periods are selected for the quality of the radiation data, and are completed by extinction profiles below the aircraft provided by a lidar. Atmospheric profiles are taken from dropsondes, and completed with ERA-Interim information. Radiative transfer simulations are performed with the LBLDIS model in an attempt to perform a radiative closure with observations. Clouds are assumed to have vertically homogeneous ice habit and effective radius, and several single scattering properties from SSPs database are used to represent cirri. It is found that none of the database used allows to match the observations in both the MIR and FIR within instrument uncertainties. It also shows that retrieving the best properties based only on MIR observations results in large residuals once applied to the FIR, and vice versa. Interestingly, any optimal set of cloud properties is characterized by an error compensation when it comes to broadband fluxes, with the MIR overestimated by the model and the FIR underestimated. The study demonstrates with unique observations the limitations of the currently used SSPs database in the FIR and point to the need for further investigation to achieve consistency between spectral regions.

The paper perfectly fits in "Atmospheric Chemistry and Physics" as it reports new findings relevant to the atmosphere energy budget and remote sensing based on a combination of novel observations and state-of-the-art radiative transfer modeling. The paper is well-written, relatively easy to follow except for a few points detailed further. While there is no doubt that the conclusions of the paper are worth being shared among the atmospheric community, I believe the paper could be significantly strengthened by extending the investigation to other SSPs models and by providing more quantitative and physical insight into the reasons for the failure of the considered database. Some leads for improvement are quickly proposed and ruled out as quickly, so that the reader, including experts in scattering, are let with few hints to how the next generation databases could perform better than the current ones. Such modifications are probably minor in terms of amount of work, but major in terms of clarification. They would make the paper more convincing and valuable to a wider audience.

**Specific comments**

1) A major issue of the paper, which might however be ruled out in a few sentences or with complementary simulations, is the treatment of the atmosphere above the aircraft. Indeed the paper clearly details how the atmosphere is prescribed below the aircraft, but nothing is said about the presence of an atmosphere above the aircraft, suggesting that deep space is considered starting at 9.3 km. At the same time the authors point in the introduction the fact that scattering is important in the FIR (l.46), much larger than in the MIR. This means that any downward flux coming from above the aircraft will be partly reflected in the FIR, hence will contribute to the observed upward radiance. The presence of a cloud above the aircraft or the tiny residual of water vapor at this altitude would certainly be visible. As a consequence the absence of any cloud should be verified and stated as long as possible, and a water vapor profile used for the whole atmosphere (for instance taking a co-located ERA-I profile). A sensitivity study could be performed to ensure that what happens above the aircraft cannot be the reason for the residuals in the FIR. Note that the additional scattering from the cloud would tend to enhance the simulated FIR radiance, which is currently underestimated.

2) The objective of the paper is to demonstrate that current SSPs databases don't work throughout the MIR and FIR spectral ranges. However to demonstrate this only one database is used, that of Baum et al. (2014). Why weren't more extensive databases used, in particular those of Yang et al. (2013) including a larger variety of habits and the effect of roughness, which is mentioned in the introduction (l.30) but not further. Also, could the database of Baran et al. (2014) be used as well? Consider also that of van Diedenhoven and Cairns (2020) to be exhaustive. If none of those databases (which probably cover all the available databases) manage to reconcile the MIR and the FIR, then the conclusion of the paper would be much stronger. At least, it should be specified to which extent the presently used database is representative of all those available in the literature.

3) The authors mention an exhaustive set of optical probes, many of them providing detailed information about the ice crystals habits and size distributions. Although it is clear that taking these information as a raw input to the simulations wouldn't work, at least because of the elapsed time between the radiative and microphysical observations, these rich observations are not mentioned at all. Maybe the complexity of the habits, the singularity of the PSDs would point to possible deficiencies of the SSPs databases. Also this could provide useful information regarding the vertical structure of the clouds, which is currently too quickly ruled out as a potential explanation for the inadequacy observed and would deserve more investigation and a dedicated sensitivity study.

4) More generally, the paper would greatly benefit from physical insight about the limitations of the databases. In which direction should experts work ? What's the next step ? Could you inform whether the temperature dependence of the refractive index may solve something. To do so it should depend on the spectral range, does it ? What about surface roughness etc. ? Such discussion could of course be very exploratory but would have the merit to provide meaningful leads for improvement.

5) What are the practical consequences of the paper for energy budgets or for ice cloud retrievals? How is cirrus radiative effect erroneous in climate simulations, how does it matter? How comes that radiative closures have been satisfying in the MIR if adding FIR channels would have resulted in different parameters? Do FIR channels provide significantly different retrievals, or do they narrow the range of possible values (hence uncertainties)?

6) Here is a suggestion to illustrate the differences in FIR and MIR retrievals for one selected case. On a 2D LUT with $r_{eff}$ and $\tau$ as axes (assuming a fixed habit), highlight the regions corresponding to MIR and FIR matching (for the different methods). This would help understand the minimisation procedure and indicate in which direction FIR channels tend to drive the retrievals (for instance).

**Technical corrections**

l.15 : single-scattering is probably more detailed than "optical" so should not be in parentheses

l.18 : state whether those fluxes are broadband or spectral

l.19 : "strong" is not quantitative, is it $\pm$ 2 or $\pm$ 10 W m$^{-2}$ ? Not clear how there can be a compensation between something that is within the residuals and something that is not.

l.22 : "cloud properties" is not defined, and the link to retrieval is not that straightforward.

l.23 : "guidance" is probably not sufficient currently for the practical development of new databases

l.26 : an additional sentence to present the SW (thin so often limited albedo) and LW (cold so large greenhouse effect) effects of cirrus clouds may be useful before talking about net effect.

l.27 : "geographical position" is not very clear. How does it impact the radiative effect? Do you mean temperature contrast with the local surface and atmosphere ? This last point should not overlap with the first two characteristics pointed out. Also, given the subsequent definition of the microphysical properties, I feel like optical thickness or particle number concentration is lacking here, unless it is included in the PSD (at its zeroth moment)

l.35 : I tend to write *in situ* as it is a Latin phrase. Holds elsewhere

l.38 : do the authors mean that all ice clouds are cirrus clouds or that they focus on cirri only ? Ice clouds could be tackled more broadly.

l.42 : no lower wavenumber limit given for the FIR ? Can be misleading

l.45 : the formulation "sensitivie to radiation" is unclear. Do you mean that the optical properties are highly variable across the FIR ? That the broadband properties are sensitive to what happens in the FIR ?

l.49 : maybe state that this holds for narrowband channels, not necessarily for hyperspectral observations

l.51 : "spectrally-resolved" has not been properly defined. Maybe give a hint to which spectral resolution this refers, because depending whether the reader is a climate modeler or a spectroscopist the expectations might differ.

l.94 : could you detail if relevant what those probes measure : PSD, scattering properties, habit ? Are all these instruments used in the paper? Are they to some extent redundant? Only relevant data should be presented.

l.96 : how are cloud phase and total amount of ice measured ?

l.98 : "information" is vague, do you mean geometric thickness here, as extinction follows ?

l.100 : here more details on the assumptions to convert backscatter profiles into extinction profiles are needed because (as discussed later on) this is key for the consistency of the synergistic radiative closure.

l.108: knowledge of the atmospheric profile above the aircraft is key as well because of scattering (including backscattering from the clouds). In particular, the absence of clouds above the aircraft is critical.

l.117: the acquisition time of a TAFTS spectrum is lacking to understand why and how 3 sets of radiance can be taken in 1 min 12 s. Please also clarify the ARIES acquisition time.

l.120: what does this "two second period" refer to? It is not clear

l.121: what's the reason for converting radiance spectra into brightness temperature (BT)? Is it practical when it comes to including instrumental error, which is more uniform in radiance than in BT?

l.125: what "variations in the cirrus properties" do you refer to? Do you simply mean the presence of cirrus ?

l.127: why is that "useful"? Is this used further in the study? Is it original, unexpected, instructive?

l.131: "a frequency dependent sensitivity to cirrus properties" is unclear. Sensitivity of what?

l.145: are these really two radiative codes, or does LBLDIS merge the LBLRTM model for gas optical thickness and DISORT for the radiative transfer equation solver?

l.148: this should be more explicit that most parameterizations try to express the single scattering properties in terms of the effective radius. Note also that it differs from the approach of Baran et al. (2014) who use temperature and ice water content to estimate single scattering properties.

l.152: this match is surprisingly low

l.153: does this emissivity model spectrally extend into the FIR?

l.154: again, no information about what the atmosphere above the aircraft looks like, although this may be critical

l.169: how many streams were used?

l.171: why only focusing on these 3 databases while Yang et al. (2013) proposes much more? In particular the effect of roughness could be investigated as a solution to overcome the current deficiencies.

l.197: separated into → composed of, split into, discretised into ?

l.218: The Baum model was already mentioned

l.227: how wide are these spectral regions?

l.231: is this "$\tau$" referring to a single cloud layer, or to the whole cloud? Is the profile still scaled on the lidar profile? Also precise whether habit and $r_{eff}$ are assumed vertically homogeneous.

l.241: could you explain what is the physical meaning of weighting by the error? What differences do you expect in comparison with the second approach? Why duplicating similar approaches?

l.242: are the Rs in the formula spectra? In which case how is the absolute difference defined? Unless one wavenumber region is actually a single channel? This should be clarified

l.244: is the minimisation performed through interpolation (or selection) of the LUT, or using a dedicated algorithm?

l.257: can such an inconsistency really explain 45% differences?

l.258: Details are needed to clarify the lidar estimate of extinction

l.260: why using two methods which are so close, in particular if the consistency is not surprising (l. 274)?

l.282: do these 14 simulations refer to method 4? Otherwise it reads like they are those among the 739 that also match FIR observations, which is obviously not the case reading the following

sentence. More explicitly, matching has not been properly defined. It is generally completed by "within uncertainties", which is clear for MIR, but not for FIR. Maybe the difference between match in the selected channels and across the whole spectrum should also be more clearly explained.

l.289-290: this suggests that no combination works in the FIR? So I guess the 14 spectra mentioned previously were matching only for the selected channels.

l.291: this is not clear why retrieval would be more constrained. If no set of parameters works, then what to conclude? Something that is sure is that using different spectral regions for the retrieval gives different results, which is of course worth pointing. But speaking of retrieval quality sounds hazardous so far.

l.295: could there be a spectral signature of the angular signal? Maybe look at spectra at 3 different viewing angles to ensure that this approximation is acceptable.

l.297: if this compensation occurs within the uncertainty range of observations, can it be considered significant? Physically, does it mean that among the possible parameters after MIR matching, FIR selects the largest/smallest $r_{eff}$ or optical thickness? This compensation should be further discussed because this provides physical insight about what individual spectral ranges would try to converge to (specific comment 6).

l.301: there is no more mention of the minimisation methods. Which method results in 2 W m$^{-2}$ errors?

l.301: I think that at this stage the main conclusion should be that none of the optical models investigated allows to match observations, which points to the need for new models. It is a result and should be mentioned before the next paragraph.

l.305: state-of-the-art for sure, but encompassing all those available in the literature?

l.308: "tested here" suggests that other models could work, so makes the conclusions weaker

l.313: how can you be sure that this tighter constraint result in a better retrieval? I think a retrieval quality should be regarded through the uncertainty associated with this retrieval, not only through the absolute error of the optimal parameters. In that sense, how does adding FIR observations help reducing the retrieval possibilities is informative.

l.314: the habit was not much discussed for the retrieval. If it is forced, are the optimal $r_{eff}$ and optical depths significantly different?

l.315: energy analysis is most meaningful at global scale. Could you provide hints to the expected global error given the distribution of cirrus (occurrence and optical depth). If 2 W m$^{-2}$ is specific to the case studied here, it could have limited implications in a climate framework.

l.320: Could you, based on your simulations, provide a more quantitative (adding a figure for instance) discussion of this potential impact on the heating rates? This would bring the attention of the climate modelers. Maybe comparing the heating/cooling rates profiles of the 4 methods for the same case.

l.324. If temperature dependence is a potential venue, could you briefly explain why this may help reconcile MIR and FIR. For this, some different sensitivities should exist in this temperature

dependence between the FIR and MIR. Is that the case? The personal communication could be expanded.

l.326: this is indeed an important point, but not sufficiently detailed. How was this vertical heterogeneity investigated? What vertical gradients were used? How could cloud probes provide quantitative information about this vertical layering? So far, the short explanation lacks details to rule out the possibility that vertical layering associated with distinct penetration depth into the cloud of the MIR and FIR (because of scattering) could be a reason for the observed mismatch. Especially when looking at the sensitivity displayed in Figure 6a.

l.331: does this mean that "new" parameterizations were built as in Baum et al. (2014) based on these new PSDs? Alike the other leads investigated, this should be quantified more properly, in terms of error bars associated with this kind of assumption of the PSDs. Other theoretical PSDs (different shapes, different widths) could also be investigated.

l.339: how do you solve the issue of concomitant cloud microphysics observation in the spaceborne configuration? Accounting for the mismatch of spatial scales.

l.340: how long is the journey to the ultimate information? Again for the modelers, the paper would benefit from providing concrete leads towards improvement. Said differently, how should a climate modeler take these results?

Table 3: the retrieved habit for the method 3 differs from all the others. Would there be an explanation why including FIR observations tends to constrain the habit to GHM?

Table 4: none of the broadband fluxes differences reaches 2 W m$^{-2}$, which seems contradictory with the statement in the text (l.301).

**References**

Baran, A. J., Hill, P., Furtado, K., Field, P., & Manners, J. (2014). A coupled cloud physics–radiation parameterization of the bulk optical properties of cirrus and its impact on the Met Office Unified Model Global Atmosphere 5.0 configuration. *Journal of Climate*, *27*(20), 7725-7752.

Baum, B. A., Yang, P., Heymsfield, A. J., Bansemer, A., Cole, B. H., Merrelli, A., ... & Wang, C. (2014). Ice cloud single-scattering property models with the full phase matrix at wavelengths from 0.2 to 100 µm. *Journal of Quantitative Spectroscopy and Radiative Transfer*, *146*, 123-139.

van Diedenhoven, B., & Cairns, B. (2020). A flexible parameterization for shortwave and longwave optical properties of ice crystals and derived bulk optical properties for climate models. *Journal of the Atmospheric Sciences*, *77*(4), 1245-1260.

Yang, P., Bi, L., Baum, B. A., Liou, K. N., Kattawar, G. W., Mishchenko, M. I., & Cole, B. (2013). Spectrally consistent scattering, absorption, and polarization properties of atmospheric ice crystals at wavelengths from 0.2 to 100 µ m. *Journal of the Atmospheric Sciences*, *70*(1), 330-347.

---

## Referee Comment (RC2) · Anonymous Referee #1 · 10 May 2020

The study presents radiative transfer simulations of airborne observations of cirrus clouds across the mid- and far-infrared (IR) wavelength region taken during the CIRC-CREX campaign. A case study is selected, where the following observations are availabe: spectral radiance observations in the mid-IR (ARIES) and in the far-IR (TAFTS), lidar observations, cloud in-situ observations, and radiosonde measurements of the humidity profile. The authors try to simultaneously fit the observations throughout a large IR spectral range (from about 100cm-1 to 1400cm-1) to radiative transfer simulations using state-of-the-art bulk optical properties of cirrus clouds by Baum et al. 2014. They find that it is not possible to find cloud microphysical properties (effective radius and crystal habit) which fit the observations throughout the whole spectrum within the

measurement uncertainties and conclude that far-IR observations can be used to better constrain retrievals of cirrus optical properties and that there is a need to develop more realistic ice cloud optical models.

The study is well presented with appropriate number of figures and well written. I recommend to publish the paper in ACP after minor revisions following the comments below.

General comments:

1. My first thought about the descrepency of retrieved optical properties in the two spectral ranges was, that the observations might be sensitive to different depths of the cloud. In the discussion, it is mentioned, that this has been investigated by varying the reff-profile within the cloud layer and it was found, that the profile has only a minor impact. I suggest to include this investigation, at least as a short appendix, rather than just mention it in one sentence in the discussion, because this is also an important result.

2. The in-situ observations are mentioned in the "Instrumentation and measurements" section but are not used because "examination of the available in-situ cloud microphysical properties [O'Shea et al., 2016] also indicated a high temporal (and therefore implied spatial) variation in the cloud PSD. These issues, combined with the knowledge that the cloud was decaying over time, suggested that it would be difficult to associate a particular observed PSD with any confidence to the radiation measurements." (p.7 l.194) - I agree that it is often difficult to compare with the in-situ observations. However, I think that you should try to at least compare the results with the in-situ observations. E.g., is the habit distribution observed in-situ similar to the general habit mixture as used by Baum et al. or is it dominated by aggregates of solid columns? The derived PSD from in-situ should also be included for comparison, even though it may not be possible to directly compare it to the results derived from the radiance observations.

Specific comments:

p.5 l.159: Where are the "present day concentrations" of CO2 and minor trace gases obtained from?

p.7 l.206: "a similar overestimate seem relative to the TAFTS measurements in the FIR micro-windows." -> I can not see this in Fig. 5, a difference plot could help.

p.8 l.235: Eq. 1 and 2: Why are absolute differences used in the fit, rather than the more commonly used quadratic differences (Chi-square fit)?

p.8 l.257: The lidar-derived value of optical thickness is smaller than that retrieved from the fit. "The deviation may be a consequence of an inconsistency between the optical properties implicitly assumed when converting the raw lidar measurements to optical depth compared with those used in the simulations here". This is a plausible explanation. Which optical properties are assumed in the lidar observation?

Fig. 7: For consistency, the transmittance should also be included in the upper panel. The transmittance curve should have a different color than "Method 1", it is particular confusing, because "Method 1" is often overplotted and not visible.

Table 4: Results from "Method 4" should be added, even though future observations restricted only to FIR are not anticipated.

---

## Author Response (AR1)

**Authors' response to all referees for ACP-2019-1181**

The authors would like to thank both referees/reviewers for their comprehensive, constructive and insightful comments and for their overall very positive reviews of this work. We have taken care to ensure that we have addressed each comment in detail, and where we have felt it appropriate to do so, we have made changes to the manuscript. As a result of the reviewers' comments and our changes, we feel this paper has been enhanced, making our findings stronger, clearer and easier to follow.

Please find below a breakdown of all referee comments (in black text) with our responses (in blue text). Where appropriate, line numbers have been included in our responses, and please note that these refer to the line numbers in the new tracked changes version of the manuscript.

**Reviewer 1 (Quentin Libois)**

1) A major issue of the paper, which might however be ruled out in a few sentences or with complementary simulations, is the treatment of the atmosphere above the aircraft. Indeed the paper clearly details how the atmosphere is prescribed below the aircraft, but nothing is said about the presence of an atmosphere above the aircraft, suggesting that deep space is considered starting at 9.3 km. At the same time the authors point in the introduction the fact that scattering is important in the FIR (l.46), much larger than in the MIR. This means that any downward flux coming from above the aircraft will be partly reflected in the FIR, hence will contribute to the observed upward radiance. The presence of a cloud above the aircraft or the tiny residual of water vapor at this altitude would certainly be visible. As a consequence the absence of any cloud should be verified and stated as long as possible, and a water vapor profile used for the whole atmosphere (for instance taking a co-located ERA-I profile). A sensitivity study could be performed to ensure that what happens above the aircraft cannot be the reason for the residuals in the FIR. Note that the additional scattering from the cloud would tend to enhance the simulated FIR radiance, which is currently underestimated.

Measurements of the downwelling radiation at the start of SLR 1 from the TAFTS instrument were used to confirm visual impressions from on-board instrument operators at the time of the flight that there were no clouds situated above the aircraft (Figure R1). It should be noted that these TAFTS downwelling observations were used for 'quick-looks' only so have not undergone the full rigorous calibration applied to the upwelling spectra analysed in the paper, which is the reason for some of the negative radiance values in the TAFTS SW channel spectrum.

The observations were compared with simulations of clear-sky downwelling radiance using the nearest ERA-I profile in space and time which, given the spatial and temporal resolution of ERA-I would also be appropriate to use to simulate downwelling radiances for the entirety of SLR1. Figure R1 shows the observations and simulations for the TAFTS SW channel. The simulations and observations show a generally excellent match, particularly in the micro-windows which would be most sensitive to the presence of cloud. In the spectral regions used for the minimisation approach described in the paper (shown by the green triangles), the downwelling radiances from the TAFTS observations are at most 2-3 mW m$^{-2}$ sr$^{-1}$ (cm$^{-1}$)$^{-1}$ (within instrument noise for this first calibration effort), while the simulated spectrum indicates lower values of almost zero. These low values, combined with the predominantly forward scattering characteristics of the ice particles, show that outside of strong water vapour lines (not used in the minimisation) there is negligible contribution to the observed upwelling radiation as a result of reflected downwelling radiation from above the aircraft.

We have added a sentence at **lines 126-127** to confirm that establishing that there was no evidence of the presence of a cloud above the aircraft at the time of the nadir radiance observations for the cases considered was also a requirement.

[Figure]

**Figure R1**: Downwelling radiance spectra at the aircraft as observed by TAFTS (black) at the start of SLR1 and simulated (red) using ERA-I profiles for T and WV (assuming a standard mid latitude winter profile for all other atmospheric components), for the SW channel.

2) The objective of the paper is to demonstrate that current SSPs databases don't work throughout the MIR and FIR spectral ranges. However to demonstrate this only one database is used, that of Baum et al. (2014). Why weren't more extensive databases used, in particular those of Yang et al. (2013) including a larger variety of habits and the effect of roughness, which is mentioned in the introduction (l.30) but not further. Also, could the database of Baran et al. (2014) be used as well? Consider also that of van Diedenhoven and Cairns (2020) to be exhaustive. If none of those databases (which probably cover all the available databases) manage to reconcile the MIR and the FIR, then the conclusion of the paper would be much stronger. At least, it should be specified to which extent the presently used database is representative of all those available in the literature.

We have amended the text to be more explicit that the databases tested and the approach used is not the only means by which cirrus radiative effects can be simulated (**lines 89-91**). However, we would point out that Baum's database does in fact include roughness and explicitly uses several of the habits modelled by Yang et al. (2013) to build the aggregate SSPs we test here, informed by extensive field campaign measurements. We agree that Baran's database would be interesting to test but it is currently being revised (Baran, personal communication, 2019) so would prefer to wait until the newer version is ready. The Van Diedenhoven and Cairns (2020) approach is a parameterisation specifically for climate models which is evaluated in part by comparison to Yang et al. (2013) and in part via comparison to the ice model used in the MODIS C6 ice cloud retrieval products (severely roughened aggregates of columns). Apart from the difficulty in including this parameterisation when the paper describing it was published after this manuscript was submitted, it seems counter-productive to use an approach which itself is evaluated via comparison to the models that already contribute to those tested here.

3) The authors mention an exhaustive set of optical probes, many of them providing detailed information about the ice crystals habits and size distributions. Although it is clear that taking these information as a raw input to the simulations wouldn't work, at least because of the elapsed time between the radiative and microphysical observations, these rich observations are not mentioned at all. Maybe the complexity of the habits, the singularity of the PSDs would point to possible deficiencies

of the SSPs databases. Also this could provide useful information regarding the vertical structure of the clouds, which is currently too quickly ruled out as a potential explanation for the inadequacy observed and would deserve more investigation and a dedicated sensitivity study.

We have addressed this as part of a more detailed response to a related question from Reviewer 2. Please see our response to Reviewer 2, General Comment 2.

4) More generally, the paper would greatly benefit from physical insight about the limitations of the databases. In which direction should experts work ? What's the next step ? Could you inform whether the temperature dependence of the refractive index may solve something. To do so it should depend on the spectral range, does it ? What about surface roughness etc. ? Such discussion could of course be very exploratory but would have the merit to provide meaningful leads for improvement.

We are observationalists at heart so in our opinion the critical next step is actually to generate a more complete observational database than the one case study analysed here. In particular we need a suite of comprehensive observations that encompasses the entire EM spectrum, with good cross-calibration where appropriate, and simultaneously measure the cloud microphysics, over a range of different cirrus cloud regimes (not simply frontal cloud as analysed here). This is explicitly stated in the paper. We do think that investigating the temperature dependence of the refractive index of ice has merit since studies have shown that the single scattering property response is more pronounced across the far infrared (greatest impact on scattering between 30 to 50 μm and absorption from 20 to 40 μm) compared to the mid infrared [Iwabuchi and Yang., 2011] with implications for ice cloud retrievals [Saito et al., 2020]. Currently, to the best of our knowledge, suitable databases incorporating this sensitivity for application to spectrally resolved measurements do not exist.

Whilst the Baum database we use already includes surface roughness, Maestri et al. (2019) noted that for thin cirrus their simulated downwelling spectra showed little sensitivity to surface roughness from smooth to severely roughened. This would suggest that this is not the major deficiency, although further observational data and associated studies would help confirm to what extent this is important.

Text has been added and amended between **lines 364-370** to reflect questions over the limitations of the current optical databases and references added accordingly.

References:
Maestri, T., C. Arosio, R. Rizzi, L. Palchetti, G. Bianchini and M. Del Guasta: Antarctic ice cloud identification and properties using downwelling spectral radiance from 100 to 1,400 cm-1, J. Geophys. Res. Atmos., 124, 4761-4781, doi:10.1029/2018/JD029205, 2019.

Saito, M., Yang, P., Huang, X., Brindley, H. E., Mlynczak, M. G. and Kahn, B. H.: Spaceborne mid- and far-infrared observations improving nighttime ice cloud property retrievals. Geophys. Res. Lett., *in press*, 2020.

5) What are the practical consequences of the paper for energy budgets or for ice cloud retrievals? How is cirrus radiative effect erroneous in climate simulations, how does it matter? How comes that radiative closures have been satisfying in the MIR if adding FIR channels would have resulted in different parameters? Do FIR channels provide significantly different retrievals, or do they narrow the range of possible values (hence uncertainties)?

We consider the first three questions here to be too big to address comprehensively in this paper. We have given a rough estimate of the longwave flux impact for this specific case but it would not be appropriate to speculate what this might be on a global scale given all the factors that influence the cloud radiative effect (as discussed in the paper's introduction). We show here that the radiative

effects are less than 1 W m$^{-2}$ when integrated from 110-1400 cm$^{-1}$, which is substantially better than the instantaneous accuracy of current broadband flux observations and hence might well be considered a satisfactory match. The main point we make is that adding the far infrared information highlights how this match is comprised of compensating effects which would not be revealed by a broadband comparison. So, given current space-based observational tools (which do not measure the spectrum across the FIR) we may not actually know how 'wrong' climate model simulations are – only new observations and comparisons can reveal this.

We also caveat again that we cannot match across the MIR and FIR simultaneously, to with the observational uncertainties, using the models tested here so our flux estimate is in some senses not representative of what the 'real' discrepancy might be, even for this single case.

6) Here is a suggestion to illustrate the differences in FIR and MIR retrievals for one selected case. On a 2D LUT with reff and τ as axes (assuming a fixed habit), highlight the regions corresponding to MIR and FIR matching (for the different methods). This would help understand the minimisation procedure and indicate in which direction FIR channels tend to drive the retrievals (for instance).

There are no simulations that simultaneously match across both regimes; this has now been emphasized (**line 317**). So it is not possible to see in which direction the FIR retrievals drive the MIR ones as the two sub-sets are independent of one another.

Technical corrections

l.15 : single-scattering is probably more detailed than "optical" so should not be in parentheses

Single-scattering has been removed to be consistent with the terminology used in the title.

l.18 : state whether those fluxes are broadband or spectral

This has been clarified by adding "spectral" (**line 19**)

l.19 : "strong" is not quantitative, is it ± 2 or ± 10 W m-2 ? Not clear how there can be a compensation between something that is within the residuals and something that is not.

"strong" has been removed. Here we are making the general point that the best performing set of optical properties (in terms of generating minimum radiance residuals) result in a compensation effect between the FIR and MIR. The implication of this is that this compensation may not be apparent if simulations are simply evaluated against broadband flux measurements, which is typical for climate models.

l.22 : "cloud properties" is not defined, and the link to retrieval is not that straightforward.

This sentence has been removed.

l.23 : "guidance" is probably not sufficient currently for the practical development of new databases

This sentence has been removed.

l.26 : an additional sentence to present the SW (thin so often limited albedo) and LW (cold so large greenhouse effect) effects of cirrus clouds may be useful before talking about net effect.

A sentence has been added to clarify the contrasting impact of clouds on incoming solar and emitted thermal radiation in the context of published results (**line 32**).

l.27 : "geographical position" is not very clear. How does it impact the radiative effect? Do you mean temperature contrast with the local surface and atmosphere ? This last point should not overlap with the first two characteristics pointed out. Also, given the subsequent definition of the microphysical properties, I feel like optical thickness or particle number concentration is lacking here, unless it is included in the PSD (at its zeroth moment)

"Geographical location" simply means latitude/longitude since, as the reviewer notes, this determines surface type. Sentence has been split in two to make the dependencies clearer, and optical thickness has been specifically mentioned (**line 36**).

l.35 : I tend to write *in situ* as it is a Latin phrase. Holds elsewhere

The formatting guidelines for the ACP Journal indicate that these should not be italicised. Therefore we have made no change.

l.38 : do the authors mean that all ice clouds are cirrus clouds or that they focus on cirri only ? Ice clouds could be tackled more broadly.

Clearly we are focusing on cirri here. However, some of the literature is more generic, and includes all ice cloud. We have revised the wording slightly in **lines 31-36** to help clarify, and then focused specifically on cirrus.

l.42 : no lower wavenumber limit given for the FIR ? Can be misleading

Actually, to the best of our knowledge there is no universally accepted lower (or even upper) bound for the FIR. It varies across communities and even within the atmospheric physics community itself. But, to be broadly consistent with the measurements we analyse here we have chosen 100-600 cm$^{-1}$. The text at **line 53** has been amended accordingly.

l.45 : the formulation "sensitivie to radiation" is unclear. Do you mean that the optical properties are highly variable across the FIR ? That the broadband properties are sensitive to what happens in the FIR ?

The text has been amended to indicate that FIR radiation is highly sensitive to the optical properties (**line 56**).

l.49 : maybe state that this holds for narrowband channels, not necessarily for hyperspectral observations

We have added "narrowband" for clarity (**line 60**).

l.51 : "spectrally-resolved" has not been properly defined. Maybe give a hint to which spectral resolution this refers, because depending whether the reader is a climate modeler or a spectroscopist the expectations might differ.

There are no global observations specifically covering the FIR whether these are hyperspectral or narrowband (or even integrated from 100-600 cm$^{-1}$) – therefore we think that defining a specific resolution here is not really helpful. We have amended the sentence to reflect this paucity (**lines 63-64**).

l.94 : could you detail if relevant what those probes measure : PSD, scattering properties, habit ? Are all these instruments used in the paper? Are they to some extent redundant? Only relevant data should be presented.

We have added this information in **line 110**. Although the data are not used directly here for the reasons explained in the paper, it is an obvious question to ask whether such data were available and so it makes sense to provide a brief summary. We have removed the detailed information on size ranges as this is not necessary.

l.96 : how are cloud phase and total amount of ice measured ?

Discussion of the cloud phase and issues with determining the phase of particles smaller than 50 μm for example, along with estimates of the ice water content derived from the different probes are covered in O'Shea *et al.*, (2016). However, to the best of our knowledge, total ice water content was not estimated from the probes. The lidar was used to estimate the ice volume extinction profiles as stated.

l.98 : "information" is vague, do you mean geometric thickness here, as extinction follows ?

Yes, this has been clarified by changing "profile" to "extent" (**line 114**).

l.100 : here more details on the assumptions to convert backscatter profiles into extinction profiles are needed because (as discussed later on) this is key for the consistency of the synergistic radiative closure.

In response to Reviewer 2, comment "p.8 l.257", we have now amended the discussion of the discrepancy between the lidar optical depth and that inferred from simulations of the observed radiances (see **lines 286-292**). We also now note (**lines 290-292**) that the required adjustment of the optical depth does not undermine the use of the relative variation of the extinction with height or the cloud thickness. Given this and considering that we do not use the optical depth from the lidar (except as a first guess) we feel that additional detail on the lidar processing is not needed here. We think that the reference provided [Fox et al., 2019] (now **line 116**) is sufficient since they provide details of the processing and a full discussion of the lidar data used here.

l.108: knowledge of the atmospheric profile above the aircraft is key as well because of scattering (including backscattering from the clouds). In particular, the absence of clouds above the aircraft is critical.

The impact of downwelling radiation from above the aircraft reflected by the cloud has been demonstrated to be negligible (see response to Reviewer 1, comment 1). We have added a sentence at **lines 126-127** to state that an additional requirement is that no cloud should be present above the aircraft.

l.117: the acquisition time of a TAFTS spectrum is lacking to understand why and how 3 sets of radiance can be taken in 1 min 12 s. Please also clarify the ARIES acquisition time.

We do not feel that the operational cycles of the interferometers are relevant to this paper since we are using the observations that are available to us given the clearly stated selection criteria. However, more details on TAFTS and ARIES can be found in Bellisario et al. [2017] and also Magurno et al. [2020].

References:

Bellisario, C. and co-authors: Retrievals of the Far Infrared Surface Emissivity Over the Greenland Plateau Using the Tropospheric Airborne Fourier Transform Spectrometer (TAFTS), JGR Atmos., doi:10.1002/2017JD027328, 2017.

Magurno, D. and co-authors: Cirrus cloud identification from airborne far-infrared and mid-infrared spectra, Remote Sens. 2020, 12(13), 2097; doi:10.3390/rs12132097, 2020.

l.120: what does this "two second period" refer to? It is not clear

This is the +/- 1 s of the TAFTS acquisition time referred to in the previous sentence.

l.121: what's the reason for converting radiance spectra into brightness temperature (BT)? Is it practical when it comes to including instrumental error, which is more uniform in radiance than in BT?

We think it is easier to show differences in brightness temperature and relate to physical properties than using the radiances. Also, yes, the uncertainties quoted for ARIES were only available expressed as a brightness temperature estimate.

l.125: what "variations in the cirrus properties" do you refer to? Do you simply mean the presence of cirrus ?

This refers to the variation between cases A, B and C which at this point we do not speculate whether this is cloud top temperature, cloud optical depth or cloud microphysical properties. Therefore, no this isn't the presence of cirrus, we are just pointing out that the cloud has different radiative properties between cases A, B and C. Therefore, we hope the existing text is appropriate and sufficient.

l.127: why is that "useful"? Is this used further in the study? Is it original, unexpected, instructive?

Useful was perhaps a bad choice of word and this has been changed to 'interesting' (**line 144**). This simply points out that the relative variation across the spectral regions, particularly where the impact of the cloud on the observed radiation is visible (i.e. in the window regions in the MIR and micro-window regions in the FIR), is not consistent for all three cases. This, we hope, suggests that the impact of the cloud on the radiation in these micro-windows is varied and complex (as we go on to discuss later in the paragraph).

l.131: "a frequency dependent sensitivity to cirrus properties" is unclear. Sensitivity of what?

A frequency dependent sensitivity of cloud's radiative signature to the cirrus properties. The text has been amended to clarify this (**lines 148, 149**).

l.145: are these really two radiative codes, or does LBLDIS merge the LBLRTM model for gas optical thickness and DISORT for the radiative transfer equation solver?

Yes these are two distinct codes, LBLRTM calculates the spectrally resolved transmission of the atmospheric layers given temperature profiles and concentrations of chosen atmospheric absorbers. This is a standalone routine that does not require LBLDIS. However, LBLRTM cannot calculate the impact of scattering from ice crystals. Therefore LBLDIS, which is essentially a routine for running the radiative transfer code, DISORT, is required to calculate the impact of the scattering and absorption of the cloud properties, combined with the spectrally resolved transmission. It does require the output from LBLRTM as you correctly point out. However, we feel that these are two distinct routines for which references have been provided.

l.148: this should be more explicit that most parameterizations try to express the single scattering properties in terms of the effective radius. Note also that it differs from the approach of Baran et al. (2014) who use temperature and ice water content to estimate single scattering properties.

We do not understand this comment. The text clearly states the simulation methodology, which follows an approach that is commonly used. Whilst we appreciate that Baran has an alternative way of relating the optical properties of an ice cloud, we are only using the optical depth and effective radius of a size distribution.

l.152: this match is surprisingly low

This is the level of agreement we see between the datasets. Perhaps the reviewer would like to elaborate why this is worthy of further comment in the paper. 'Low' is quite hard to interpret.

l.153: does this emissivity model spectrally extend into the FIR?

No, the longest wavelength is 13 μm (~769 cm$^{-1}$). A sentence has been added to clarify that the Masuda model only applies for wavenumbers down to ~769 cm$^{-1}$ (**line 170**), and a spectrally invariant value of 0.99 was used for lower frequencies (**lines 171-172**). However, the opaque nature of the atmosphere in the FIR between the surface and cirrus cloud means that there is no impact of the FIR surface emissivity on the aircraft measured radiances (**lines 172-173**).

l.154: again, no information about what the atmosphere above the aircraft looks like, although this may be critical

Please see our response to your comment "1)".

l.169: how many streams were used?

16 – this information has been added in **line 190**.

l.171: why only focusing on these 3 databases while Yang et al. (2013) proposes much more? In particular the effect of roughness could be investigated as a solution to overcome the current deficiencies.

Please see our response to your comment "2)"

l.197: separated into → composed of, split into, discretised into ?

"separated into" has been changed to "split into" (**line 220**).

l.218: The Baum model was already mentioned

Indeed, but this refers to the particular Baum model – the Aggregate Solid Columns (ASC).

l.227: how wide are these spectral regions?

A single channel is ~ 2 cm$^{-1}$ in each spectral region to avoid a dilution in the sensitivity of the minimisation to strong absorption features not attributed to the cloud properties (e.g. water vapour). The channel width is now clarified (**line 256**).

l.231: is this "τ" referring to a single cloud layer, or to the whole cloud? Is the profile still scaled on the lidar profile? Also precise whether habit and reff are assumed vertically homogeneous.

Optical depth refers to the entire cloud, the profile was scaled using the lidar profile and r$_{eff}$ was assumed vertically homogeneous. An extra sentence has been added to the figure caption for "Table 2" to indicate this and there is also substantial extra discussion in section 4.2 to address a comment from Reviewer 2.

l.241: could you explain what is the physical meaning of weighting by the error? What differences do you expect in comparison with the second approach? Why duplicating similar approaches?

Weighting the differences by the uncertainty allows the relative importance of the difference compared to the uncertainty in each minimisation region to be taken into account. This is similar to a Chi Square approach. Following a comment from Reviewer 2 "p.8 l.235", the minimisation method of

Eq. 1 was repeated using a Chi Square. The same results were obtained for the Chi Square approach as when using Eq.1. The second approach simply looks at the total error assuming that all uncertainties are equal across the spectrum.

Since the approach used to determine the differences does not fundamentally impact the conclusion, that no single simulated spectrum can match the observations across the entire mid and far-infrared, we feel it is helpful to retain both approaches.

l.242: are the Rs in the formula spectra? In which case how is the absolute difference defined? Unless one wavenumber region is actually a single channel? This should be clarified

One wavenumber region is a single channel ($\sim 2$ cm$^{-1}$ wide). This is clarified by the new wording in **lines 255-256** which now does not use 'region'.

l.244: is the minimisation performed through interpolation (or selection) of the LUT, or using a dedicated algorithm?

Selection. The discretisation of the parameters (e.g. optical depth, effective radius of the particle size distribution) was chosen to be sufficient that the observational uncertainties were much greater than the quantised variability of the simulated radiative spectra.

l.257: can such an inconsistency really explain 45% differences?

It is our understanding that the primary source of the uncertainty remains in the selection of the lidar ratio to produce a volume extinction coefficient. However, since the lidar extinction profiles are simply used to constrain the vertical extent of the cloud and provide an estimate of the relative IWC profile within the cloud, we do not seek to account for these differences. We feel this is beyond the scope of this paper. However, please also see response to Reviewer 2, "p.8 l.257" and updates in **lines 286-292**.

l.258: Details are needed to clarify the lidar estimate of extinction

Please see response to Reviewer 2, "p.8 l.257".

l.260: why using two methods which are so close, in particular if the consistency is not surprising (l.274)?

This indicates that the variation in the estimated uncertainties in the observed spectral radiances is not a key influence on the result. However, we feel it is useful to present both methods. Also see response to your comment "l.241".

l.282: do these 14 simulations refer to method 4? Otherwise it reads like they are those among the 739 that also match FIR observations, which is obviously not the case reading the following sentence. More explicitly, matching has not been properly defined. It is generally completed by "within uncertainties", which is clear for MIR, but not for FIR. Maybe the difference between match in the selected channels and across the whole spectrum should also be more clearly explained.

Yes. Additional words have been added to clarify this point (**line 317**). This was actually stated in the sentence following the original l282 but this has now been removed as it would simply repeat the information contained in the revised **line 317**.

l.289-290: this suggests that no combination works in the FIR? So I guess the 14 spectra mentioned previously were matching only for the selected channels.

Yes, this is hopefully clear following the amendment to answer the previous comment (**line 317**).

l.291: this is not clear why retrieval would be more constrained. If no set of parameters works, then what to conclude? Something that is sure is that using different spectral regions for the retrieval gives different results, which is of course worth pointing. But speaking of retrieval quality sounds hazardous so far.

We agree, this could be misinterpreted. We have replaced the sentence with one that more accurately captures our intent – to state that the combination of FIR and MIR measurements can provide a more rigorous test of the ability of ice cloud optical property models to correctly capture the cloud radiative signature than MIR observations alone (**lines 326-327**).

l.295: could there be a spectral signature of the angular signal? Maybe look at spectra at 3 different viewing angles to ensure that this approximation is acceptable.

This figure is provided as rough estimate only as stated in the text. A more thorough treatment would be required to definitively answer this point, possibly using more than 3 viewing angles. In addition, simulations at angles other than nadir would also make the assumption that the angular scattering of the cirrus is correctly (or at least consistently well) captured by the optical models across the spectral range sampled, which might be unlikely given the results shown here.

l.297: if this compensation occurs within the uncertainty range of observations, can it be considered significant? Physically, does it mean that among the possible parameters after MIR matching, FIR selects the largest/smallest reff or optical thickness? This compensation should be further discussed because this provides physical insight about what individual spectral ranges would try to converge to (specific comment 6).

We reiterate: there is no combination of $r_{eff}$ and optical depth that allows us to simultaneously match across the MIR and FIR within the uncertainties. Please also see our response to your comment 6.

l.301: there is no more mention of the minimisation methods. Which method results in 2 W m-2 errors?

Hopefully this is clear that now that Table 4 includes results from all four approaches used (in response to a comment from Reviewer 2). The text also indicates that method 3 (and now method 4) provides discrepancies that exceed 2 W m$^{-2}$ in **lines 339-340**.

l.301: I think that at this stage the main conclusion should be that none of the optical models investigated allows to match observations, which points to the need for new models. It is a result and should be mentioned before the next paragraph.

We do not understand the need to repeat this message here.

l.305: state-of-the-art for sure, but encompassing all those available in the literature?

We do not use all available in the literature, please see response to your comment "2)" for further details. However the statement clearly refers to the Baum optical models, which themselves have been shown to represent the optical properties of ice clouds in the MIR based on an extensive series of field campaigns.

l.308: "tested here" suggests that other models could work, so makes the conclusions weaker

As you note in your comment "2)" we have not tested all models so therefore feel this wording is appropriate.

l.313: how can you be sure that this tighter constraint result in a better retrieval? I think a retrieval quality should be regarded through the uncertainty associated with this retrieval, not only through the absolute error of the optimal parameters. In that sense, how does adding FIR observations help reducing the retrieval possibilities is informative.

This is an interesting question, but we feel it is one that is beyond the scope of this paper. We do not actually say anything about retrievals here.

l.314: the habit was not much discussed for the retrieval. If it is forced, are the optimal reff and optical depths significantly different?

We would like to reemphasize that we are not performing a retrieval. We are looking for the simulation that most closely matches the observations. However, to answer your question, we re-ran the minimisation using approach 1, forcing the habit to be confined to the ASC for Case A. The closest matching spectrum related to a simulation using an increased optical depth of ~0.06 (compared with unconstrained habit presented in the results), and the effective radius of the PSD increased from 34 to 40 μm. The corresponding absolute flux differences for the three bands (MIR, SW FIR and LW FIR – i.e. the 3 columns from table 4) using approach 1 were: 1.11 (1.10), 1.18 (-0.64), 0.25 (-0.08) Wm$^{-2}$. Therefore fixing the habit does of course affect the choice of simulated spectrum most closely matching the observation, but this increases the differences (as would be expected), and hence is worse. We hope this answers your question, but we do not think this is relevant information to include in the paper.

l.315: energy analysis is most meaningful at global scale. Could you provide hints to the expected global error given the distribution of cirrus (occurrence and optical depth). If 2 W m-2 is specific to the case studied here, it could have limited implications in a climate framework.

Not necessarily for all applications but we agree the general point from a climate perspective. However, it is impossible to give a hint on what the global error would be from this single case study given all the factors that influence cirrus radiative effect as discussed in the introduction, not least the optical thickness.

l.320: Could you, based on your simulations, provide a more quantitative (adding a figure for instance) discussion of this potential impact on the heating rates? This would bring the attention of the climate modelers. Maybe comparing the heating/cooling rates profiles of the 4 methods for the same case.

We think this is beyond the scope of this paper. We hope that the pointer we provide in the discussion might motivate further study into the potential effects on vertical heating rates. This same comment also applies to your previous point ("l.315").

l.324. If temperature dependence is a potential venue, could you briefly explain why this may help reconcile MIR and FIR. For this, some different sensitivities should exist in this temperature dependence between the FIR and MIR. Is that the case? The personal communication could be expanded.

We now point towards the paper by Iwabuchi and Yang [2011] which demonstrates the impact of this temperature dependence spectrally, and to some new work by Saito et al. [2020] (**lines 364 to 368**).

l.326: this is indeed an important point, but not sufficiently detailed. How was this vertical heterogeneity investigated? What vertical gradients were used? How could cloud probes provide quantitative information about this vertical layering? So far, the short explanation lacks details to rule out the possibility that vertical layering associated with distinct penetration depth into the cloud of

the MIR and FIR (because of scattering) could be a reason for the observed mismatch. Especially when looking at the sensitivity displayed in Figure 6a.

Please see our response to Reviewer 2, General Comment 1.

l.331: does this mean that "new" parameterizations were built as in Baum et al. (2014) based on these new PSDs? Alike the other leads investigated, this should be quantified more properly, in terms of error bars associated with this kind of assumption of the PSDs. Other theoretical PSDs (different shapes, different widths) could also be investigated.

Yes, using Ping Yang's individual optical models (Baum's are a hybrid of these), new PSDs were generated. However, for reasons already mentioned, the observations are poorly constrained given the time difference and variability of the cloud reported by the lidar and indeed the variability seen within the in-situ data itself. For this reason we do not see the value of providing a detailed description of these studies. Please also see our response to Reviewer 2, General Comment 2.

l.339: how do you solve the issue of concomitant cloud microphysics observation in the spaceborne configuration? Accounting for the mismatch of spatial scales.

Whilst this is an important question, we do not think this is something that this paper should be asked to address. In order to answer this, it would require an entirely new study in its own right, however, we can say that dedicated under-flights with suitable instrumentation will obviously be required.

l.340: how long is the journey to the ultimate information? Again for the modelers, the paper would benefit from providing concrete leads towards improvement. Said differently, how should a climate modeler take these results?

The goal of this paper is to make researchers, including climate modellers, understand that there remain significant uncertainties in representing the optical properties of ice clouds consistently across the infrared and that these uncertainties can propagate to sizeable radiative effects that might not be manifested in broadband comparisons. We think the paper conveys this message.

Table 3: the retrieved habit for the method 3 differs from all the others. Would there be an explanation why including FIR observations tends to constrain the habit to GHM?

The most likely explanation is the enhanced sensitivity to habit in the far-infrared (see Fig. 6b) compared to a relatively flat response in the mid-infrared.

Table 4: none of the broadband fluxes differences reaches 2 W m-2, which seems contradictory with the statement in the text (l.301).

Broadband refers to the 3 "broadband channels" of LW FIR (110 to 300 cm$^{-1}$), SW FIR (320 to 540 cm$^{-1}$) and MIR (600 to 1400 cm$^{-1}$). Therefore Table 4 shows that for approach #3 for case B, there is a negative difference of -2.02. However, as a result of a request by Reviewer 2 to include all 4 approaches, there are now many more examples of this value reaching and exceeding 2 Wm$^{-2}$.

**Reviewer 2 (Anonymous Referee #1)**

General comments:

1. My first thought about the descrepency of retrieved optical properties in the two spectral ranges was, that the observations might be sensitive to different depths of the cloud. In the discussion, it is

mentioned, that this has been investigated by varying the reff-profile within the cloud layer and it was found, that the profile has only a minor impact. I suggest to include this investigation, at least as a short appendix, rather than just mention it in one sentence in the discussion, because this is also an important result.

This has now been added in **Section 4.2**. Your comment motivated a re-examination of the sensitivity which has improved Figure 6 such that the baseline simulation is now consistent across all four perturbation experiments (this was not previously the case). Because of this the results show larger sensitivity to a perturbation in the vertical profile of $r_{eff}$ than was originally reported but this would not be distinguishable from a vertically uniform perturbation in $r_{eff}$ and has a much smaller magnitude (compare Fig. 6(a) to 6(d)). For this reason we continue to assume $r_{eff}$ is vertically uniform in our approach since incorporating vertical variation would not change our overall conclusion regarding the FIR-MIR inconsistency.

The text has been updated in Section 4.2 to account for the addition of Fig. 6(d), and also the other changes made. A new Fig.6 has been produced and the figure caption updated accordingly.

2. The in-situ observations are mentioned in the "Instrumentation and measurements" section but are not used because "examination of the available in-situ cloud microphysical properties [O'Shea et al., 2016] also indicated a high temporal (and therefore implied spatial) variation in the cloud PSD. These issues, combined with the knowledge that the cloud was decaying over time, suggested that it would be difficult to associate a particular observed PSD with any confidence to the radiation measurements." (p.7 l.194) - I agree that it is often difficult to compare with the in-situ observations. However, I think that you should try to at least compare the results with the in-situ observations. E.g., is the habit distribution observed in-situ similar to the general habit mixture as used by Baum et al. or is it dominated by aggregates of solid columns?

The derived PSD from in-situ should also be included for comparison, even though it may not be possible to directly compare it to the results derived from the radiance observations.

We think it is important to point out that we are not trying to hide anything by not including the in-situ results, simply that the rapid evolution of the cloud system being studied coupled with the 50 minute (to over two hours) delay between the radiative observations and the in-situ observations makes the comparison pointless since we really aren't sampling the same cloud. Indeed, as we now mention in the text (**Section 4.1**) the in-situ measurements themselves show a rapid variation over the period that they were collected. However, for information, we include below the relevant plots from O'Shea et al. [2016]. These are derived from the in-situ observations, binned according to cloud temperature (a proxy for altitude). These indicate that there was no consistently dominant habit, with perhaps the exception of the coldest layer sampled within the cloud, during the in-situ measurements (Figure 6 (b)).

[Figure]

**Figure 6.** Proportions of particle habits observed by the CPI within different temperature regions of the clouds for (a) 11 March 2015 and (b) 13 March 2015. The temperatures correspond to the same runs/profiles shown in Figure 5.

They also suggest a variable PSD which may have a temperature-size dependence (Figure 5(b)) but this was difficult to quantify given the uncertainties surrounding the measurements of the smaller ice particles made by the 2DS (O'Shea, Pers. Comm. 2019), further questioning the value of examining the PSD data for this study. However, very recent work, soon to be submitted for publication may improve the understanding of the 2DS data and its comparison with the HALOHolo (O'Shea, Pers. Comm. 2020).

[Figure]

**Figure 5.** (continued)

**O'Shea et al. [2016] Figure 5 (adapted). Particle size distributions from the 2DS (black lines) and CIP 100 (red lines) probes for different temperature regions for (b) 13 March 2015. HALOHolo observations are shown in blue. The PSDs have been averaged over individual runs/profiles made by the FAAM BAe-146.**

Figures 5(b) and 6 (b) shown above have been reproduced from O'Shea et al. [2016] with permission from the author. The original figures are available online https://agupubs.onlinelibrary.wiley.com/doi/full/10.1002/2016JD025278 (open access).

**Specific comments**

p.5 l.159: Where are the "present day concentrations" of CO2 and minor trace gases obtained from?

These were obtained from the NOAA-ESRL measurements from Mace Head, a reference has been added at **lines 179-180**.

 p.7 l.206: "a similar overestimate seem relative to the TAFTS measurements in the FIR micro-windows." -> I can not see this in Fig. 5, a difference plot could help.

Closer examination of the differences in fact show that the differences in the MIR are around 4 K and only around 2 K in FIR. The text at **line 229** has been amended to indicate this. We have included a difference plot here (Fig. R2) for completeness but do not feel it adds much to the paper, since the differences between the observations and simulations are examined in much greater detail in Fig. 7.

[Figure]

Figure R2: Simulation minus observation relating to the observed and simulated (cloudy-sky) spectra shown in Fig. 5.

p.8 l.235: Eq. 1 and 2: Why are absolute differences used in the fit, rather than the more commonly used quadratic differences (Chi-square fit)?

Equation 1 weights the differences by the uncertainty, allowing the relative importance of the difference compared to the uncertainty in each minimisation region to be taken into account. This is similar to a Chi Square approach. To check, this was repeated using a Chi Square, and the same results were obtained to those using Eq.1.

The second approach (Eq. 2) simply looks at the total error implicitly assuming that all uncertainties are equal across the spectrum. This was included as an alternative way to identify the best matching simulations to the observations. Irrespective of the approach used, there is no impact on the conclusion, that no single simulated spectrum can match the observations across the entire mid and far-infrared.

p.8 l.257: The lidar-derived value of optical thickness is smaller than that retrieved from the fit. "The deviation may be a consequence of an inconsistency between the optical properties implicitly assumed when converting the raw lidar measurements to optical depth compared with those used in the simulations here". This is a plausible explanation. Which optical properties are assumed in the lidar observation?

We have rephrased the discussion of the lidar optical depth (**lines 286-292**) to better reflect the way the lidar data are processed, since the lidar optical depth is derived from the lidar extinction which is a function of the lidar extinction-to-backscatter ratio, commonly referred to as the lidar ratio. This ratio is a function of the optical properties of the cloud but is determined directly from the lidar dataset rather than being derived via a cloud microphysical model. A more detailed explanation follows.

In this study we used the Baum models to enable wavelength interpolation of the lidar optical depth determined at 355 nm. The lidar optical depth was determined from the lidar derived volume extinction coefficient at 355 nm and the lidar observed cloud thickness. The lidar data were analysed to obtain profiles of both the extinction coefficient and backscatter. This was carried out using a constant value of the backscatter to extinction ratio (lidar ratio) of 25 sr [Fox et al., 2019] which is considered typical for cirrus [e.g. Young et al., 2013]. However, the value of this ratio is expected to vary with the details of the cloud microphysics, and both theoretical [e.g. Ding et al., 2016] and observational [e.g. Chen et al., 2002; Gouveia et al., 2017] studies indicate that variations of roughly +/- 50% around of this value are not unusual.

Therefore, adjustment of the lidar ratio well within this plausible range would produce a lidar optical depth consistent with the values obtained from the simulations (i.e the closest matching simulation to the observations). Although strictly speaking applying such an adjustment within the lidar processing would induce subtly different corrections within the profile, over the extinction coefficient range here this effect is expected to be negligible compared to other sources of measurement error. Hence in this analysis, we simply use the lidar data as processed, to provide information on the cloud geometrical thickness and the relative variation of extinction within the cloud (and for a "first guess" of the cloud optical depth). We then scale the derived optical depth, noting that the scaling required to provide agreement between the lidar and the simulations is not outside of what might be expected.

We have clarified in the text that the scaling of the lidar optical depth is analogous to adjusting the assumed backscatter-to-extinction (lidar) ratio from the mean value used. We note that this is within the plausible range for cirrus, and this would not significantly impact the lidar determined relative variation of the extinction within the cloud or its geometrical extent.

Fig. 7: For consistency, the transmittance should also be included in the upper panel. The transmittance curve should have a different color than "Method 1", it is particular confusing, because "Method 1" is often overplotted and not visible.

The transmittance has been added to the upper panel and the colour has been changed.

Table 4: Results from "Method 4" should be added, even though future observations restricted only to FIR are not anticipated.

These results have now been added.

**A test of the ability of current bulk optical models to represent the radiative properties of cirrus cloud across the mid- and far-infrared**

Richard J. Bantges[1,2], Helen E. Brindley[1,2], Jonathan E. Murray[2], Alan E. Last[2], Jacqueline E. Russell[2], Cathryn Fox[3], Stuart Fox[3], Chawn Harlow[3], Sebastian J. O'Shea[4], Keith N. Bower[4], Bryan A. Baum[5], Ping Yang[6], Hilke Oetjen[7] and Juliet C. Pickering[2]

[1]National Centre for Earth Observation, Imperial College London, UK
[2]Physics Department, Imperial College London, UK
[3]Met Office, UK
[4]University of Manchester, UK
[5]Science and Technology Corporation, Madison, USA
[6]Department of Atmospheric Sciences, Texas A&M University, USA
[7]ESA/ESTEC, Noordwijk, Netherlands

*Correspondence to*: Richard J. Bantges (r.bantges@imperial.ac.uk)

**Abstract.** Measurements of mid- to far-infrared nadir radiances obtained from the UK Facility for Airborne Atmospheric Measurements (FAAM) BAe-146 aircraft during the Cirrus Coupled Cloud-Radiation Experiment (CIRCCREX) are used to assess the performance of various ice cloud bulk optical  property models. Through use of a minimisation approach, we find that the simulations can reproduce the observed spectra in the mid-infrared to within measurement uncertainty but are unable to simultaneously match the observations over the far-infrared frequency range. When both mid and far-infrared observations are used to minimise residuals, first order estimates of the spectral flux differences between the best performing simulations and observations indicate a  compensation effect between the mid- and far--infrared such that the absolute broadband difference is < 0.7 W m$^{-2}$. However, simply matching the spectra using the mid-infrared (far-infrared) observations in isolation leads to substantially larger discrepancies, with absolute differences reaching ~ 1.8 (3.1) W m$^{-2}$. These results show that simulations using these microphysical models may give a broadly correct integrated longwave radiative impact but that this masks spectral errors, with implicit consequences for the vertical distribution of atmospheric heating. They also imply that retrievals using these models applied to mid-infrared radiances in isolation will select cirrus optical properties that are inconsistent with far-infrared radiances. As such they highlight the potential benefit of more extensive far-infrared observations for the assessment and, where necessary, the improvement of current ice bulk  optical models.

**1 Introduction**

The role of ice clouds (e.g. cirrus) in determining the radiative balance of the Earth and its atmosphere is particularly complex and uncertain [e.g. Baran et al., 2014b; Yang et al., 2015]. Recent calculations based on ice cloud properties retrieved from

active satellite instruments suggest that ice clouds have a net warming effect on the global climate, with enhanced trapping of outgoing infrared radiation, particularly from cirrus anvils, exceeding enhanced reflection of incoming solar radiation [Hong

35    et al., 2016]. More generally, the net radiative effect of all types of ice cloud, including cirrus, is critically dependent upon its optical thickness, which itself is linked to the cloud microphysical properties. Other important characteristics include  the vertical position and extent, and the geographical location of the cloud [e.g. Heymsfield et al., 2013; Hong and Liu, 2015]. Key microphysical parameters include ice particle habit, particle size distribution (PSD), and morphology such as aggregation, roughness and concavity [Zhang et al., 1999;

[revised manuscript text omitted]

**4.2 Achieving an improved simulation-observation fit**

It is evident from Fig. 5 that the initial choice of cloud parameters used to simulate the observed radiance were sub-optimal. The two key parameters typically required to define the microphysical properties of an ice cloud are $r_{eff}$ and the ice particle habit. The sensitivity of the mid-infrared to ice particle size has been known for many years [e.g. Bantges et al., 1999] and more recently sensitivity studies extending into the far-infrared have been performed [e.g. Yang, 2003; Yang et al., 2013]. Figure 6 (a-c) shows the results of a series of simulations performed to examine the impact of varying $r_{eff}$, ice particle habit and optical depth for case A.; the simulated radiances are developed using the Baum et al. [2014] models. As part of this analysis, while we have no direct observational evidence for a variation in $r_{eff}$ with height through the cloud it is reasonable to explore the response to such a variation since it might be expected to be different in the MIR and FIR regimes. Hence Figure 6(d) shows the impact of varying $r_{eff}$ in three 1 km thick layers through the cloud.

The resultsFigure 6 (a) indicates that there are a wide range of spectral regions that demonstrate sensitivity to size from approximately 300 to 500-600 cm$^{-1}$, 750 to 850 cm$^{-1}$ and 950 to 1250 cm$^{-1}$. Note however that the ordering of the differences changes around 400 cm$^{-1}$ where the largest $r_{eff}$ no longer shows the greatest sensitivity. In contrast, there are spectral regions that exhibit sensitivity primarily to ice particle habit (Fig. 6(b)). Differences between the ASC GHM and SC model are greatest around 550-500 cm$^{-1}$, while they are greatest at around 400 cm$^{-1}$ for ASC GHM and GHM ASC differences. Sensitivity to increasing optical depth (Fig. 6(c)) is broadly similar across the MIR and FIR from 400 to 1400 cm$^{-1}$, but drops off rapidly at wavenumbers lower than 400 cm$^{-1}$ due to the increasing effect of strong water vapour absorption and the overall reduction in radiative energy at these frequencies. Introducing a vertical variation in $r_{eff}$ (Fig. 6(d)) produces a change which has a very similar spectral shape, but much smaller magnitude, to that observed for a bulk change in $r_{eff}$.

With use ofUsing this information, a scheme was developed to minimise the differences between the simulated and observed

255   spectra in regions showing particular sensitivity to $r_{eff}$ and habit. Four wavenumber regions channels in the MIR and four in

the FIR were used; 775, 850, 900 and 1200 cm$^{-1}$ in the MIR and 365, 410, 450 and 497 cm$^{-1}$ in the FIR, each of ~ 2 cm$^{-1}$ width.

The ultimate goal was to investigate whether there was any combination of parameters that could fit the observations across

the MIR and FIR simultaneously given measurement uncertainties. Because of the reduced sensitivity to the vertical profile of

$r_{eff}$ and the similarity of the associated spectral signature to those generated from vertically uniform perturbations to $r_{eff}$ we do

260   not attempt to account for any vertical variation in our simulations.

[revised manuscript text omitted]

365 show that the spectral effects of this temperature dependence on the ice optical properties has the greatest impact on the scattering between ~ 200 to 330 $cm^{-1}$ and on the absorption between 250 and 500 $cm^{-1}$. Studies imply that this  can exert a noticeable impact on retrievals of cirrus optical depth and effective radius utilising the  mid- and  far-infrared simultaneously [Saito et al., 2020]. Other parameters included in the optical property databases, such as ice particle roughness, which although observed to be less important than habit and size [Maestri et al.

370 2019], may also require consideration in future. A further question relates to the relative sensitivity of the far- and mid-infrared regimes to depth within the cloud. To address this, the impact of varying the vertical profile of $r_{eff}$ within the cloud on the simulated spectra was also considered , but was found to have an almost identical spectral signature to that generated by a bulk change in $r_{eff}$ but of a greatly reduced magnitude..

[revised manuscript text omitted]

**Table 4: Integrated, simulation minus observation, flux differences for selected spectral ranges (MIR, SW FIR and LW FIR) and the total, for the best matching spectra obtained using the four different minimisation methods, for the three case studies. Results from method 4 are  included for completeness  although we do not anticipate future observations to be restricted to only FIR frequencies.**